# ALAS: Additive Learnable Alpha-Stable Kernels for Flexible Bayesian Optimization

Weibo Huang [1]   Cheng Hua [1]

## Abstract

Bayesian Optimization is widely used for expensive black-box optimization, yet its success often depends on choosing a kernel that matches the objective's unknown structure. In this work, we propose ALAS, a flexible Gaussian Process kernel family built from symmetric $\alpha$-stable spectral components. By learning the stability parameter $\alpha$, ALAS adapts its effective smoothness from data, capturing both smooth trends and sharp irregularities. We present two parameterizations: ALAS, a single stationary component with joint spectral modulation, and ALAS-Sep, a separable variant that learns dimension-wise tail behavior to improve robustness on approximately decomposable objectives. Experiments on standard benchmarks and real-world surrogates demonstrate strong and robust performance across diverse settings.

## 1. Introduction

Black-box optimization under tight evaluation budgets is pervasive across science and engineering. Bayesian optimization (BO) addresses this setting by fitting a Gaussian process (GP) surrogate and selecting evaluations with an acquisition function. In BO, the GP kernel is the primary source of inductive bias and often dominates sample efficiency. However, widely used stationary kernels (SE/Matérn/RQ/periodic) impose a fixed notion of smoothness, which can underfit objectives with multi-scale variation, sharp local irregularities, or long-range dependence.

A principled way to enrich stationary kernels is to learn their spectral density through Bochner's theorem (Stein, 1999). Spectral mixture (SM) kernels (Wilson & Adams, 2013)

model the spectral density $S(\omega)$ as a Gaussian mixture, offering closed-form covariances and strong empirical results. While follow-up variants have further expanded this flexibility for BO (Remes et al., 2017), two challenges remain. First, Gaussian spectral components are light-tailed in frequency, so capturing heavy-tailed or strongly low-frequency structures may require many components. Second, as the dimension grows, these models are harder to fit reliably, and their effective complexity is harder to characterize, making BO analysis based on eigenvalue decay and information gain less transparent.

**This work.** We propose ALAS, a learnable $\alpha$-stable kernel family. ALAS is built from a symmetric $\alpha$-stable (S$\alpha$S) spectral component, where the stability parameter $\alpha \in (0, 2]$ directly controls spectral tail heaviness. By learning $\alpha$ from data via GP marginal likelihood, ALAS smoothly interpolates from Gaussian-like behavior ($\alpha = 2$) to heavier-tailed regimes ($\alpha < 2$), including the Cauchy landmark ($\alpha = 1$). This allows the surrogate to represent both smooth trends and sharp variations without manually choosing a fixed smoothness kernel class. To better handle higher-dimensional inputs, we introduce ALAS-Sep, a separable construction that sums one-dimensional ALAS components across coordinates. ALAS-Sep enables dimension-wise adaptivity: each coordinate learns its own tail parameter $\alpha_j$, while the learned component scales reflect dimensional relevance. On the theory side, we link spectral tails to Mercer eigenvalue decay, yielding explicit information-gain and regret rates controlled by $\alpha$. In particular, at the Gaussian boundary we recover the classical poly-logarithmic information gain in one dimension.

We summarize the main contributions of this paper as follows:

**Learnable $\alpha$-stable kernel family.** We propose ALAS, a stationary GP kernel family based on symmetric $\alpha$-stable spectral components. Learning the stability parameter $\alpha$ provides a data-driven way to adjust spectral tail behavior and effective regularity, ranging from near-Gaussian ($\alpha = 2$) to heavier-tailed regimes ($\alpha < 2$), and capturing both smooth structure and sharp variations within a single kernel family.

[1]Antai College of Economics and Management, Shanghai Jiao Tong University, Shanghai, China. Correspondence to: Weibo Huang <3338725059@sjtu.edu.cn>, Cheng Hua <cheng.hua@sjtu.edu.cn>.

*Proceedings of the $43^{rd}$ International Conference on Machine Learning*, Seoul, South Korea. PMLR 306, 2026. Copyright 2026 by the author(s).

**Separable variant with theory.** We introduce ALAS-Sep, a separable construction that sums one-dimensional ALAS components. This structured parameterization improves robustness in multi-dimensional BO by letting each dimension learn its own $\alpha_j$ while retaining exact GP inference. We further connect spectral tail behavior to Mercer eigenvalue decay and derive explicit information-gain and regret bounds for the one-dimensional component and the separable variant.

**Empirical validation.** We evaluate ALAS on a diverse set of synthetic and real-world benchmarks ranging from $d = 3$ to 30. The results suggest complementary strengths: ALAS is a robust default on low-to-moderate dimensional tasks with strong interactions, while ALAS-Sep is most effective on larger-dimensional and approximately decomposable objectives by learning dimension-specific tail parameters. Across these settings, ALAS achieves strong and robust performance against competitive baselines.

## 2. Related Work

**Stationary Priors in BO** Bayesian Optimization (BO) often models the objective function with a stationary GP prior. Common choices include the squared exponential (SE), Matérn, or Rational Quadratic (RQ) kernels, often combined with Automatic Relevance Determination (ARD) to capture different length scales (Williams & Rasmussen, 2006; Srinivas et al., 2009; Snoek et al., 2012; Shahriari et al., 2015). These kernels work well for reasonably smooth, single-scale objectives, but they impose a fixed smoothness assumption and a specific form of spectral decay. As a result, they can be inefficient when the true objective exhibits multi-scale structure, sharp local variations, or long-range dependencies. Two main directions have been explored to improve flexibility. The first learns a more flexible input representation or covariance structure, for example through deep kernel learning, deep GPs, input warping, or non-stationary kernels (Wilson et al., 2016; Damianou & Lawrence, 2013; Snoek et al., 2014; Paciorek & Schervish, 2003). The second, which we follow, keeps the stationary kernel structure but learns the spectral density directly using Bochner's Theorem. This includes spectral mixture kernels and their variants, which have been explored recently in BO (Wilson & Adams, 2013; Remes et al., 2017). Our work extends this spectral approach by introducing learnable $\alpha$-stable spectral components, and a separable variant (ALAS-Sep) for improved robustness in multi-dimensional BO.

**Spectral Kernel Learning.** Spectral kernel learning stems from Bochner's theorem, which represents any stationary kernel as the Fourier transform of a spectral density $S(\omega)$ (Stein, 1999). The spectral mixture (SM) kernel approximates this density with Gaussian mixtures, allowing it to capture complex patterns with closed-form covariances (Wilson & Adams, 2013). Related approaches explore generalized spectral shapes (Samo & Roberts, 2015) or sparse approximations to improve efficiency (Lázaro-Gredilla et al., 2010; Jang et al., 2017). However, a key limitation remains: Gaussian spectral components are light-tailed in frequency, so matching rough or heavy-tailed behaviors can require many mixture components.

**Heavy-Tailed Priors.** A common approach to improve robustness is to replace the GP prior with heavier-tailed stochastic processes, such as Student-$t$ processes (Shah et al., 2014). Recent work also studies heavy-tailed priors in Bayesian neural networks and nonparametric models (Fortuin et al., 2021; Loría & Bhadra, 2023; Agapiou & Castillo, 2024). Previous work also studies $\alpha$-stable processes, where the stability parameter $\alpha \in (0, 2]$ directly governs tail heaviness (Samorodnitsky & Taqqu, 1994). This family includes two well-known landmarks: $\alpha = 2$ recovers the Gaussian case, while $\alpha = 1$ recovers the Cauchy case. In the kernel setting, the Cauchy case corresponds to a Lorentzian spectrum and an exponential stationary covariance (Nolan, 2020; Rahimi & Recht, 2007). Our approach is distinct: we keep the GP framework to preserve tractable inference, but model the kernel spectrum with $\alpha$-stable components and learn $\alpha$ by marginal likelihood, so the kernel adapts its effective regularity from data.

**Eigen-decay, information gain, and BO theory.** GP regret bounds are typically expressed in terms of the maximum information gain $\gamma_T$, which is controlled by the decay of the kernel's Mercer eigenvalues (Srinivas et al., 2012; Chowdhury & Gopalan, 2017). For stationary kernels, the decay of eigenvalues reflects the tail behavior of the spectral density. Specifically, Gaussian spectra lead to rapid eigenvalue decay, whereas polynomial spectral tails result in slower decay (Widom, 1963). This motivates our design: while heavier spectral tails can increase theoretical complexity, our separable construction (ALAS-Sep) reduces this cost by decomposing the kernel across dimensions (Kandasamy et al., 2015). We derive explicit regret bounds for ALAS. In particular, at the Gaussian boundary $\alpha = 2$, we recover the standard poly-logarithmic rates.

## 3. Preliminaries

We study Bayesian optimization (BO) for maximizing an unknown objective $f : \mathcal{X} \subset \mathbb{R}^d \rightarrow \mathbb{R}$ over a compact domain $\mathcal{X}$. We quantify performance by the cumulative regret $R_T := \sum_{t=1}^{T} (f(x^\star) - f(x_t))$, where $x^\star \in \arg\max_{x \in \mathcal{X}} f(x)$.

## 3.1. Gaussian Processes

A Gaussian process (GP) prior $f \sim \mathcal{GP}(0, k)$ is specified by a covariance kernel $k$. Given noisy observations $\mathcal{D}_n = \{(x_i, y_i)\}_{i=1}^n$ with $y_i = f(x_i) + \varepsilon_i$ and $\varepsilon_i \sim \mathcal{N}(0, \sigma^2)$, the posterior at $x$ is Gaussian with:

$$\mu_n(x) = \mathbf{k}_n(x)^\top (\mathbf{K}_n + \sigma^2 \mathbf{I})^{-1} \mathbf{y}_n, \tag{1}$$

$$\sigma_n^2(x) = k(x, x) - \mathbf{k}_n(x)^\top (\mathbf{K}_n + \sigma^2 \mathbf{I})^{-1} \mathbf{k}_n(x), \tag{2}$$

where $\mathbf{K}_n$ is the kernel matrix on training data. Kernel hyperparameters are learned by maximizing the log marginal likelihood (MLL):

$$\log p(\mathbf{y}_n \mid \mathbf{X}_n, \theta) = -\frac{1}{2} \mathbf{y}_n^\top (\mathbf{K}_n + \sigma^2 \mathbf{I})^{-1} \mathbf{y}_n \tag{3}$$

$$-\frac{1}{2} \log |\mathbf{K}_n + \sigma^2 \mathbf{I}| + C. \tag{4}$$

## 3.2. Spectral Representations of Stationary Kernels

A kernel is stationary if $k(\mathbf{x}, \mathbf{x}') = k(\tau)$ depends only on $\tau = \mathbf{x} - \mathbf{x}'$. By Bochner's theorem (Stein, 1999), stationary kernels are characterized by nonnegative spectral measures.

**Theorem 3.1** (Bochner's Theorem). *A continuous function $k(\tau)$ on $\mathbb{R}^d$ is positive definite if and only if it is the Fourier transform of a finite non-negative measure $\mu$ on $\mathbb{R}^d$:*

$$k(\tau) = \int_{\mathbb{R}^d} e^{i 2\pi \omega^\top \tau} d\mu(\omega). \tag{5}$$

*When the spectral measure admits a density, we write $d\mu(\omega) = S(\omega) d\omega$ and call $S(\omega)$ the spectral density.*

This duality maps kernel design to spectral density modeling. The tail behavior of $S(\omega)$ as $\|\omega\| \to \infty$ governs the effective smoothness of GP sample paths. Two examples are:

- Gaussian Decay: $S(\omega) \propto e^{-c\|\omega\|^2}$. This implies that sample paths are infinitely smooth, filtering out high-frequency variations.

- Polynomial Decay: $S(\omega) \propto (1 + \|\omega\|^2)^{-\nu - d/2}$. This allows for modeling finitely smooth functions, where the roughness is controlled by $\nu$.

Existing methods (Wilson & Adams, 2013) approximate $S(\omega)$ using Gaussian mixtures, which inherit a super-exponential decay rate. However, this implies infinite sample path smoothness and often struggles to capture rough features or heavy-tailed spectral behaviors.

## 3.3. Mercer Spectrum and Information Gain

Our regret analysis follows the standard GP-UCB framework, where the key complexity measure is the maximum information gain $\gamma_T$. To connect the spectral tail of a stationary kernel to $\gamma_T$, we upper bound $\gamma_T$ via the Mercer eigenvalue decay of the associated kernel integral operator on the compact domain $\mathcal{X}$.

Let $\nu$ be a finite measure on $\mathcal{X}$. Since $\mathcal{X}$ is compact and $k$ is continuous, the associated integral operator $T_k : L^2(\mathcal{X}, \nu) \to L^2(\mathcal{X}, \nu)$,

$$(T_k g)(x) = \int_{\mathcal{X}} k(x, z) g(z) d\nu(z),$$

is compact and admits a discrete nonnegative spectrum.

**Theorem 3.2** (Mercer's Theorem). *There exist eigenvalues $\{\lambda_h\}_{h=1}^\infty$ with $\lambda_h \geq 0$ and orthonormal eigenfunctions $\{\phi_h\}_{h=1}^\infty \subset L^2(\mathcal{X}, \nu)$ such that for all $x, x' \in \mathcal{X}$,*

$$k(x, x') = \sum_{h=1}^\infty \lambda_h \phi_h(x) \phi_h(x'). \tag{6}$$

**Maximum information gain.** For noisy GP observations with variance $\sigma^2$, the mutual information between $f$ and observations at a design set $A = \{x_1, \dots, x_T\} \subset \mathcal{X}$ is

$$I(\mathbf{y}_A; f_A) = \frac{1}{2} \log \det \left( I + \sigma^{-2} K_A \right),$$

where $K_A$ is the $T \times T$ kernel matrix on $A$. The maximum information gain is

$$\gamma_T := \max_{A \subset \mathcal{X} : |A| = T} I(\mathbf{y}_A; f_A). \tag{7}$$

**Spectral control via Mercer eigenvalues.** A standard consequence of Theorem 3.2 is that $\gamma_T$ can be upper bounded by the leading Mercer eigenvalues (Srinivas et al., 2012; Vakili et al., 2021):

$$\gamma_T \leq \frac{1}{2} \sum_{h=1}^T \log \left( 1 + \sigma^{-2} T \lambda_h \right). \tag{8}$$

This uses the standard comparison $\lambda_i(K_A) \leq T \lambda_i(T_k)$ for any $A$ with $|A| = T$. Therefore, bounding $\gamma_T$ reduces to understanding the eigenvalue decay $\{\lambda_h\}$, which we later relate to the tail behavior of the stationary spectral density $S(\omega)$.

## 4. Method

We introduce the Additive Learnable Alpha-Stable (ALAS) kernel family, a stationary Gaussian process (GP) parameterization that treats the stability index $\alpha \in (0, 2]$ as a learnable hyperparameter. Optimizing $\alpha$ directly adjusts the spectral tail, continuously interpolating between smooth Gaussian-like ($\alpha = 2$) and rough heavy-tailed ($\alpha < 2$) behaviors. We first define ALAS as a single $d$-dimensional

stationary kernel with dimension-wise scales, whose $d=1$ specialization recovers the one-dimensional ALAS component in (12). To improve optimization stability and enable dimension-wise adaptivity in higher dimensions, we further introduce a separable variant, ALAS-Sep, which sums one-dimensional ALAS components across coordinates so that each dimension can learn its own effective roughness $\alpha_j$. Both constructions retain exact GP inference and are trained by marginal likelihood maximization, providing a principled alternative to manual kernel selection.

### 4.1. Spectral Representation and Tail Behavior

We first focus on one-dimensional stationary kernels $k(\tau)$ with $\tau = x - x'$ and use the $2\pi$-Fourier convention

$$k(\tau) = \int_{\mathbb{R}} e^{i2\pi\omega\tau} S(\omega)\, d\omega, \qquad S(\omega) \geq 0. \quad (9)$$

Equivalently, if we view $S$ as a probability density function (PDF), then $k(\tau)$ is the corresponding characteristic function evaluated at $t = 2\pi\tau$.

**A S$\alpha$S spectral component.** Let $p(\omega; \alpha, \delta)$ denote the density of a symmetric $\alpha$-stable distribution with $\alpha \in (0, 2]$ and scale $\delta > 0$. Since $p(\omega; \alpha, \delta)$ lacks a closed-form expression for general $\alpha$, its characteristic function $\varphi(t)$ is defined:

$$\varphi(t) := \int_{\mathbb{R}} e^{it\omega}\, p(\omega; \alpha, \delta)\, d\omega = \exp\big(-|\delta t|^\alpha\big). \quad (10)$$

Substituting $S(\omega) = p(\omega; \alpha, \delta)$ into (9) yields the closed-form base kernel:

$$\begin{aligned}
\kappa_{\alpha,\delta}(\tau) &= \varphi(2\pi\tau) \\
&= \exp\big(-(2\pi\delta|\tau|)^\alpha\big) = \exp\Big(-\Big|\frac{\tau}{\ell}\Big|^\alpha\Big),
\end{aligned} \quad (11)$$

where $\ell := (2\pi\delta)^{-1}$ is the characteristic lengthscale. The range $\alpha \in (0, 2]$ is the standard domain for symmetric $\alpha$-stable laws, with $\alpha = 2$ corresponding to the Gaussian case (Nolan, 2020), and it is also the usual positive-definite range of the powered-exponential envelope.

**Relation to powered-exponential kernels.** The base envelope $\kappa_{\alpha,\delta}$ in (11) is the classical powered-exponential covariance family (Diggle et al., 1998). ALAS extends this envelope with a learned modulation frequency $\gamma$, which captures oscillatory and non-monotone correlations. In this spectral parameterization, $\alpha$ controls tail decay, while $\gamma$ controls the dominant frequency.

**The modulated 1D ALAS component.** We define a single one-dimensional ALAS component by modulating the powered-exponential envelope:

$$k_{\text{ALAS}}^{(1)}(\tau) = w \cdot \kappa_{\alpha,\delta}(\tau) \cdot \cos(2\pi\gamma\tau), \quad (12)$$

where $w > 0$ is a component weight, and the parameters $\theta = \{w, \alpha, \delta, \gamma\}$ are learned jointly by marginal likelihood maximization.

**Tail heaviness.** For $\alpha \in (0, 2)$, the symmetric $\alpha$-stable density has polynomial tails, $p(\omega; \alpha, \delta) \asymp |\omega|^{-(1+\alpha)}$ as $|\omega| \to \infty$, which retains substantial high-frequency mass. At the Gaussian boundary $\alpha = 2$, $p(\omega; 2, \delta)$ is Gaussian and decays as $\exp(-c\omega^2)$ for some $c > 0$, which suppresses high frequencies much more strongly.

**Shifted spectrum under modulation.** The cosine modulation in (12) introduces a characteristic frequency $\gamma$. By the modulation property of the Fourier transform, multiplying by $\cos(2\pi\gamma\tau)$ corresponds to shifting the base spectrum to a symmetric pair:

$$S_{\text{ALAS}}^{(1)}(\omega) = \frac{w}{2}\Big[p(\omega - \gamma; \alpha, \delta) + p(\omega + \gamma; \alpha, \delta)\Big]. \quad (13)$$

The factor $1/2$ comes from the cosine decomposition, so the total component weight remains $w$. This shift changes where spectral mass concentrates but does not change the tail exponent. Hence, for $\alpha \in (0, 2)$ we still have $S_{\text{ALAS}}^{(1)}(\omega) \asymp |\omega|^{-(1+\alpha)}$ as $|\omega| \to \infty$, so $\alpha$ remains the explicit knob controlling tail heaviness and effective roughness.

### 4.2. Learning Complexity: Eigenvalues and Regret

The spectral tail determines the effective capacity of the kernel, which we formalize through Mercer eigenvalue decay and the resulting information-gain bounds.

**Proposition 4.1** (Eigenvalue decay). *Consider the one-dimensional ALAS component $k_{\text{ALAS}}^{(1)}$ defined in (12) on a compact interval $\mathcal{X} = [0, L] \subset \mathbb{R}$, with the Mercer operator on $L_2(\mathcal{X})$. Assume $\alpha \in (0, 2)$. Then the Mercer eigenvalues $\{\lambda_h\}_{h \geq 1}$ satisfy*

$$\lambda_h = \Theta\Big(h^{-(1+\alpha)}\Big). \quad (14)$$

This polynomial eigen-decay implies a polynomial bound on the maximum information gain, which then yields the standard GP-UCB regret bound (Srinivas et al., 2009).

**Theorem 4.2** (Information gain and regret). *Under the conditions of Proposition 4.1, the maximum information gain after $T$ queries satisfies*

$$\gamma_T = \tilde{\mathcal{O}}\Big(T^{\frac{1}{1+\alpha}}\Big). \quad (15)$$

*Consequently, the cumulative regret satisfies*

$$R_T = \tilde{\mathcal{O}}\Big(\sqrt{T\gamma_T}\Big) = \tilde{\mathcal{O}}\Big(T^{\frac{1}{2} + \frac{1}{2(1+\alpha)}}\Big), \quad (16)$$

*where $\tilde{\mathcal{O}}(\cdot)$ hides poly-logarithmic factors. Moreover at $\alpha = 2$, the 1D information gain becomes poly-logarithmic:*

$$\gamma_T = \mathcal{O}\big((\log T)^2\big), \qquad R_T = \mathcal{O}\big(\sqrt{T}\log T\big). \quad (17)$$

Theorem 4.2 highlights $\alpha$ as a complexity knob:

- Larger $\alpha$ corresponds to lighter spectral tails and faster eigenvalue decay, leading to smaller information-gain bounds; at the Gaussian boundary $\alpha = 2$, $\gamma_T$ is poly-logarithmic in 1D.

- As $\alpha \to 0$, the spectral tail becomes heavier, eigenvalues decay more slowly, and $\gamma_T$ grows polynomially, reflecting higher sample complexity for sharp variations.

- Unlike kernels with a fixed smoothness choice, ALAS learns $\alpha$ from data by marginal likelihood, adapting between near-Gaussian behavior on smooth objectives and heavier-tailed spectra when sharp variations are present.

### 4.3. Multi-dimensional Inputs

In BO with $d$-dimensional inputs and tight evaluation budgets, learning a flexible stationary kernel is statistically challenging, since the model must infer structure across many directions from sparse data.

Here we focus on statistical scalability rather than reducing the $\mathcal{O}(n^3)$ cost of exact GP inference. We introduce two constructions: ALAS, a single $d$-dimensional component and ALAS-Sep, an additive construction with separated per-dimension parameters.

**ALAS**   We define ALAS as a single $d$-dimensional stationary component with dimension-wise scales and a joint modulation frequency. Let $\tau = x - x' \in \mathbb{R}^d$. We set

$$k_{\text{ALAS}}(x, x') = w \exp\left(-\sum_{j=1}^{d}(2\pi\delta_j|\tau_j|)^\alpha\right)\cos\left(2\pi\,\gamma^\top\tau\right),$$
(18)

where $w > 0$, $\delta_j > 0$, $\gamma \in \mathbb{R}^d$, and $\alpha \in (0, 2]$ is shared across dimensions. When $\alpha = 2$ and $\gamma = 0$, (18) reduces to the ARD squared-exponential kernel. This formulation is expressive through the joint modulation, while remaining a single shared stability parameter $\alpha$ across all dimensions.

**ALAS-Sep.**   To impose a stronger structural prior under limited data, we adopt an additive decomposition (Kandasamy et al., 2015; Gardner et al., 2017; Rolland et al., 2018), which improves learnability by modeling the objective as a sum of low-dimensional effects. We define

$$k_{\text{ALAS}}^{\text{sep}}(x, x') = \sum_{j=1}^{d} k_{\text{ALAS}}^{(j)}(x_j - x_j'; \theta_j),$$
(19)

with coordinate-wise one-dimensional components

$$k_{\text{ALAS}}^{(j)}(\tau; \theta_j) = w_j\,\kappa_{\alpha_j,\delta_j}(\tau)\cos(2\pi\gamma_j\tau),$$
(20)

where $\theta_j = \{w_j, \alpha_j, \delta_j, \gamma_j\}$ and $\kappa_{\alpha_j,\delta_j}$ is defined in (11). This construction separates hyperparameters across dimensions and allows each coordinate to learn its own tail parameter $\alpha_j$.

**Information gain under additive prior.**   A theoretical benefit of (19) is that its spectrum decomposes across 1D components, which yields an information-gain bound linear in $d$ (see Appendix A.3).

**Lemma 4.3** (Additive information gain). *For an additive kernel $k = \sum_{j=1}^{d} k_j$, the maximum information gain satisfies*

$$\gamma_T(k) \;\leq\; \sum_{j=1}^{d}\gamma_T(k_j).$$
(21)

Let $\alpha_{\min} := \min_{j\in[d]}\alpha_j$. Applying Lemma 4.3 and Theorem 4.2 to each one-dimensional component gives

$$\gamma_T\left(k_{\text{ALAS}}^{\text{sep}}\right) \;\leq\; \sum_{j=1}^{d}\tilde{\mathcal{O}}\left(T^{\frac{1}{1+\alpha_j}}\right) \;\leq\; \tilde{\mathcal{O}}\left(d \cdot T^{\frac{1}{1+\alpha_{\min}}}\right).$$
(22)

Thus, the dimension dependence of information gain enters only through the linear factor $d$. Moreover, the overall rate is governed by the least smooth coordinate, the one with the smallest $\alpha_j$. Together with the standard GP-UCB reduction $R_T = \tilde{\mathcal{O}}(\sqrt{T\,\gamma_T})$, (22) yields sublinear regret and preserves the same $T$-dependence as the one-dimensional bound, up to the factor $\sqrt{d}$.

**Comparison to existing kernel classes.**   ALAS augments the powered-exponential kernel with a learnable tail parameter $\alpha$ and modulation $\gamma$, so it can adapt smoothness from data rather than fixing a kernel class. Compared with spectral mixture kernels based on Gaussian components, ALAS uses an $\alpha$-stable spectral component with a learnable tail exponent to directly control spectral tail behavior and function roughness. Compared with heavy-tailed process priors such as Student-$t$ processes, ALAS keeps standard GP inference and expresses heavy-tailed behavior through the kernel spectrum. (Park & Baek, 2001; Wilson & Adams, 2013; Shah et al., 2014)

**Practical learning.**   In implementation, we fit an exact GP by maximizing the marginal log-likelihood. We optimize an unconstrained variable and map it to $\alpha \in (0, 2]$ through a smooth transform, so the constraint is always satisfied. Weak priors on the noise and scale parameters help prevent degenerate fits when $n$ is small. We initialize $(\gamma, \delta)$ from an empirical spectrum heuristic and initialize $\alpha$ near 2, letting marginal likelihood move it toward rougher regimes when supported by data.

Algorithm 1 follows the standard BO loop; the only change

**Algorithm 1** Bayesian Optimization with ALAS

---

**Input:** search space $\mathcal{X} \subset \mathbb{R}^d$, budget $T$, initial size $n_0$, acquisition function $acq(\cdot)$.

**Initialize:** choose an ALAS kernel $k_{\text{ALAS}}(\cdot, \cdot; \Theta)$; place a GP prior $f \sim \mathcal{GP}(0, k_{\text{ALAS}}(\cdot, \cdot; \Theta))$; collect $\mathcal{D}_{n_0} = \{(x_i, y_i)\}_{i=1}^{n_0}$.

1: **for** $t = n_0, \ldots, T - 1$ **do**
2:    **Hyperparameter learning**
3:    Fit $\hat{\Theta}_t$ by maximizing marginal log-likelihood on $\mathcal{D}_t$
4:    Compute posterior mean $\mu_t(\cdot)$ and variance $\sigma_t^2(\cdot)$ under $\hat{\Theta}_t$
5:    **Acquisition and query**
6:    $x_{t+1} \leftarrow \arg\max_{x \in \mathcal{X}} acq(x; \mu_t, \sigma_t)$
7:    Query the black-box objective and observe $y_{t+1} = f(x_{t+1}) + \varepsilon$; update $\mathcal{D}_{t+1} \leftarrow \mathcal{D}_t \cup \{(x_{t+1}, y_{t+1})\}$
8: **end for**
9: **return** best observed point $x_{\text{best}}$ and the learned stability parameters $\hat{\alpha}$ (or $\{\hat{\alpha}_j\}$ for ALAS-Sep).

---

is that the surrogate uses an ALAS kernel whose hyperparameters $\Theta$ are re-estimated by marginal likelihood at each iteration. Here $\Theta = \{\alpha, \gamma, \delta, w\}$ includes both stability and spectral parameters. The two kernel constructions (ALAS and ALAS-Sep) share the same loop and differ only in how $k_{\text{ALAS}}(\cdot, \cdot; \Theta)$ is parameterized, see Algorithm 2 in Appendix for details.

## 5. Experiments

### 5.1. Adaptivity to Function Smoothness

A key motivation of the ALAS Kernel is adaptivity: it should be flexible enough to fit irregular objectives, yet automatically recover standard Gaussian behavior on smooth data. We illustrate this behavior with two controlled one-dimensional experiments, comparing the 1D ALAS component with an RBF kernel under identical training data and likelihood settings.

**Scenario 1: Fitting Roughness.** We first consider the Weierstrass function, an example known for being everywhere continuous yet nowhere differentiable. As shown in Fig. 1, the RBF posterior mean assumes excessive smoothness and fails to capture the narrow valley near the global minimum. In contrast, ALAS adapts by learning a smaller $\alpha$ and can model local singularities better. This yields a tighter fit around high-curvature regions and improved predictive log-likelihood (PLL) on the test grid.

**Scenario 2: Preserving Smoothness** We next test whether ALAS becomes unnecessarily complex when the ground truth is smooth by sampling data from an RBF GP prior. As shown in Fig. 2, even when initialized with a

heavier-tailed setting ($\alpha = 1.5$), ALAS converges towards $\hat{\alpha} = 2.0$ and recovers an RBF-like posterior. The posterior means and uncertainty are similar, indicating that ALAS does not overfit and remains competitive on standard smooth functions.

**Remark.** These two extremes illustrate the dual capability of the ALAS Kernel: it shifts toward heavy-tailed behavior when the objective is rough, and converges towards the Gaussian limit when the objective is smooth.

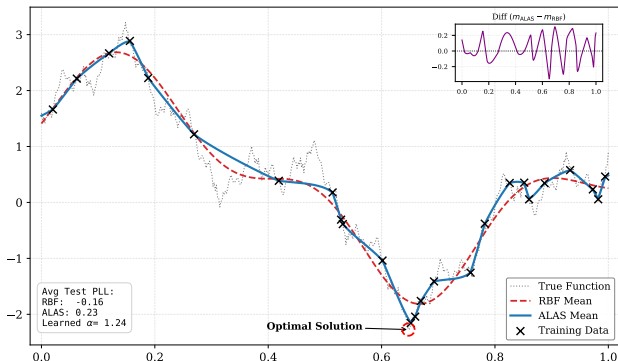

*Figure 1.* **Learning roughness beyond Gaussian kernels.** We fit two GPs on the same Weierstrass training set ($n = 25$): an RBF kernel and the 1D ALAS component $k_{\text{ALAS}}^{(1)}$. We report the average predictive log-likelihood (PLL) on a dense test grid. For this representative run, the 1D ALAS component achieves a higher PLL and learns $\hat{\alpha} = 1.24$. See App. B.4 for seed-sweep statistics.

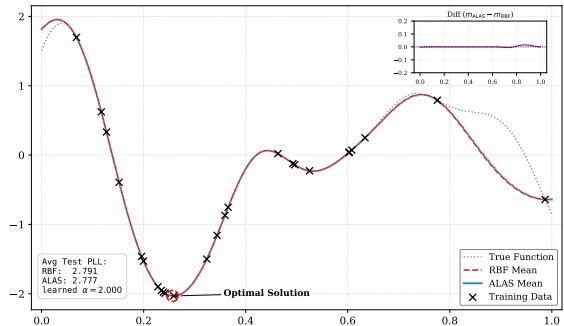

*Figure 2.* **Recovering Gaussian Smoothness.** On a smooth function ($n = 25$), the 1D ALAS component automatically adapts to the simple landscape by learning $\hat{\alpha} = 2.0$. Consequently, its posterior mean becomes virtually indistinguishable from the RBF baseline, showing that ALAS recovers Gaussian behavior on smooth data.

### 5.2. Optimization Performance

We evaluate the proposed ALAS framework on a diverse set of synthetic benchmarks and real-world problems, ranging from $d = 3$ to $d = 30$. We compare two variants of our framework: ALAS, a single $d$-dimensional stationary kernel with ARD scaling and one shared tail parameter, and ALAS-Sep, which sums per-dimension ALAS components and

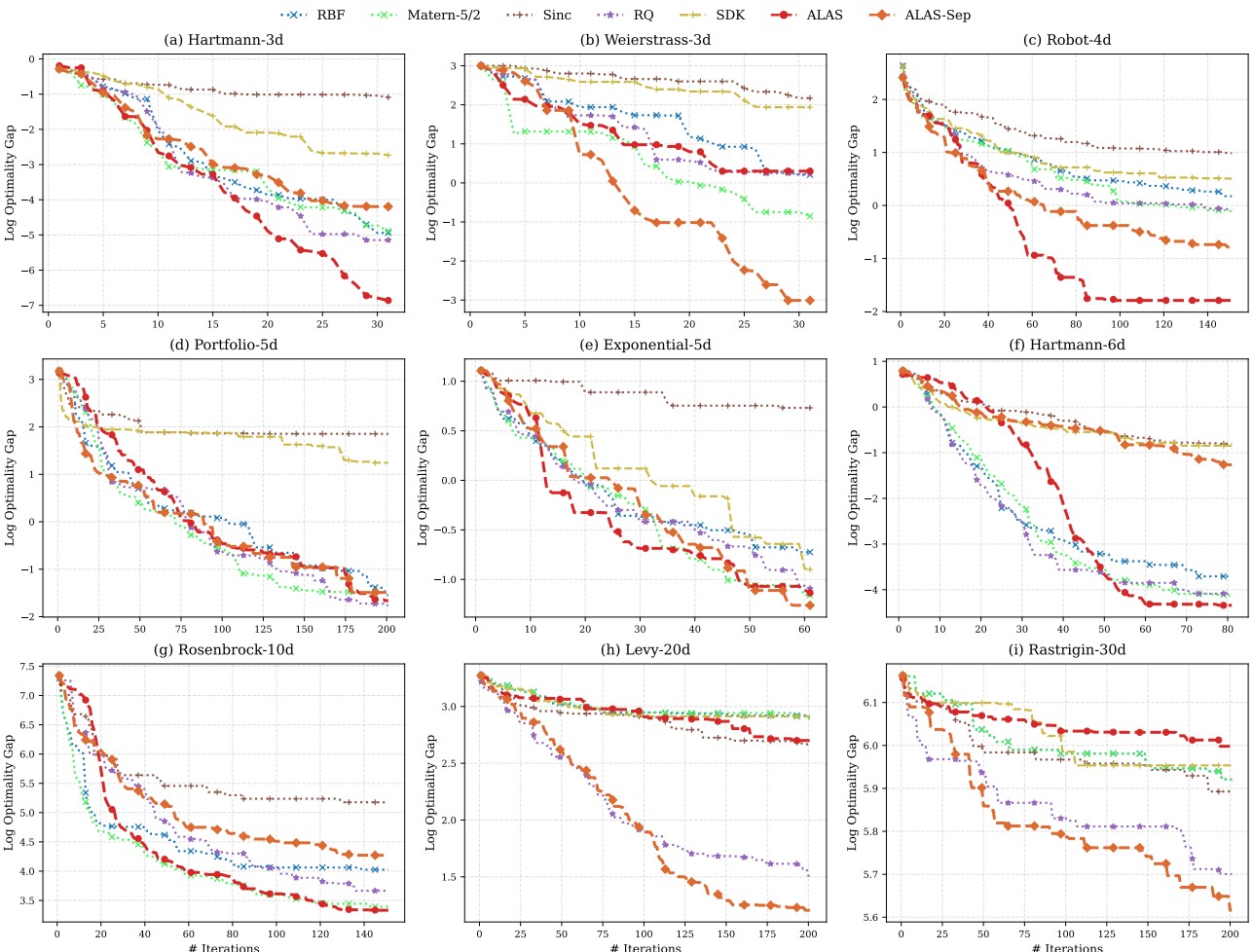

*Figure 3.* **Optimization performance on benchmarks.** We report the best-so-far log optimality gap over BO iterations (mean over 10 seeds) on tasks with $d \in [3, 30]$ using Expected Improvement (EI). Lower is better. ALAS (red) remains competitive across tasks and is particularly strong on interaction-heavy or irregular objectives. ALAS-Sep (orange) shows clear gains on approximately decomposable or oscillatory landscapes, while remaining competitive elsewhere.

learns dimension-wise tail parameters. Baselines include standard stationary kernels (RBF, Matérn-$5/2$, RQ) and spectral kernels (Sinc, SDK) (Vargas-Hernández & Gardner, 2021; Tobar, 2019). All methods share the same BO pipeline and optimization settings (see details in Appendix B.1).

Figure 3 summarizes the results using Expected Improvement (EI). To provide a consistent measure of convergence across all tasks, we report the log optimality gap. For synthetic benchmarks and the Robot Pushing task, the global optimum is known. For the Portfolio Optimization task, we estimate the global maximum based on extensive offline evaluations of the surrogate model.

**ALAS as a robust default.** Across smoother benchmarks, ALAS achieves strong performance, indicating that learning $\alpha$ does not sacrifice performance on regular objectives. In these cases, $\alpha$ naturally converges toward the Gaussian

boundary ($\alpha = 2$), effectively recovering a smooth spectral mixture kernel.

On tasks with strong feature interactions, such as Rosenbrock and the two real-world problems, ALAS often outperforms standard baselines and its separable counterpart. This suggests that a single $d$-dimensional component can better capture cross-dimensional dependencies.

We also observe a "warm-up" effect in several tasks: ALAS may improve more slowly in the first few iterations than fixed-smoothness baselines. This is expected, as learning the additional parameter $\alpha$ requires data. However, ALAS often catches up and overtakes later as the hyperparameters stabilize and better match the objective's structure.

**When separability helps.** ALAS-Sep is often strongest when the objective is closer to a sum of low-dimensional ef-

*Table 1.* Learned effective stability parameter for the ALAS family. For ALAS-Sep, $\alpha$ denotes the average of the dimension-wise $\alpha_j$ values. $\alpha_{\text{range}}$ gives the 10th–90th percentiles over the aggregated iteration history.

| Benchmark ($d$) | $\alpha_{\text{FINAL}}$ | $\alpha_{\text{RANGE}}$ (10–90%) |
|---|---|---|
| *High final $\alpha$* | | |
| Exponential (5) | $1.96 \pm 0.03$ | $[1.60, 2.00]$ |
| Hartmann (6) | $1.92 \pm 0.03$ | $[1.86, 1.95]$ |
| Rastrigin (30) | $1.91 \pm 0.03$ | $[1.50, 2.00]$ |
| Levy (20) | $1.89 \pm 0.04$ | $[1.60, 2.00]$ |
| Hartmann (3) | $1.86 \pm 0.06$ | $[1.85, 2.00]$ |
| Rosenbrock (10) | $1.83 \pm 0.04$ | $[1.77, 1.99]$ |
| *Lower final $\alpha$* | | |
| Robot (4) | $1.82 \pm 0.21$ | $[1.02, 2.00]$ |
| Portfolio (5) | $1.67 \pm 0.12$ | $[1.48, 1.91]$ |
| Weierstrass (3) | $1.56 \pm 0.34$ | $[1.26, 2.00]$ |

fects, or when different dimensions exhibit different degrees of roughness (e.g., Weierstrass-3d, Levy-20d, and Rastrigin-30d). By learning a separate $\alpha_j$ per dimension, it can model sharp variation in some coordinates while keeping others smoother, which is harder to do with a single shared $\alpha$.

**Interpreting the learned $\alpha$.** A key feature of the ALAS family is its ability to adapt effective regularity by learning stability parameters from data. To monitor this behavior, we record the learned $\alpha$ at every BO iteration. For ALAS-Sep, we summarize the per-iteration stability parameter by $\bar{\alpha} := \frac{1}{d} \sum_{j=1}^{d} \alpha_j$ (for ALAS this equals the single learned $\alpha$). Table 1 reports $\alpha_{\text{final}}$ as the mean $\pm$ std of $\bar{\alpha}_T$ across seeds, and $\alpha_{\text{range}}$ as the 10th–90th percentiles of the pooled values $\{\bar{\alpha}_t\}$ over all seeds and BO iterations. We only use these statistics for interpretation, not for tuning or selection. Appendix B.4 provides an additional 1D interpretation with known Fourier-coefficient decay rates.

Overall, the learned $\alpha$ is consistent with the empirical regularity of the objectives. On smoother benchmarks such as Exponential and Hartmann, ALAS typically converges towards the Gaussian boundary $\alpha = 2$. On more irregular objectives, such as Weierstrass and Portfolio, it often selects a smaller $\alpha$ with higher variability. On mixed landscapes like Levy and Rastrigin, it tends to converge to intermediate values.

In practice, ALAS is a safer choice when cross-dimensional interactions are expected, while ALAS-Sep is preferable when the objective is approximately additive or dimensions may have different roughness.

## 6. Conclusion

In this work, we introduce ALAS, a stationary GP kernel family that learns a tail parameter $\alpha$ to adapt its effective regularity from data. To better handle multi-dimensional

inputs under limited evaluations, we also propose ALAS-Sep, a separable construction that sums one-dimensional ALAS components and learns dimension-specific tail parameters. The two parameterizations are complementary: ALAS is suited to objectives with cross-dimensional interactions, while ALAS-Sep is useful when the objective is approximately additive or dimensions differ in roughness. We support this design with theory linking spectral tails to Mercer eigenvalue decay and information gain, and with empirical results showing robust performance across diverse benchmarks.

While ALAS improves statistical efficiency, reducing the cubic computational cost with scalable GP approximations remains an important next step. On the modeling side, a useful direction is to study intermediate structures between shared-$\alpha$ ALAS and fully per-dimension ALAS-Sep. Sparse interactions or low-dimensional variable groups are natural candidates for higher-dimensional BO under tight budgets.

## Acknowledgements

We thank Yi Zhang for helpful discussions on spectral mixture kernels and for sharing an earlier implementation. C. Hua is partly supported by the National Natural Science Foundation of China (72301172, 72394370:72394375, 72495130:72495132) and the Shanghai Jiao Tong University Office of Liberal Arts (ZHWK2502).

## Impact Statement

This work studies kernel design for Bayesian optimization through learnable spectral tail behavior, providing theoretical guarantees that connect spectral decay, information gain, and regret. The results offer a principled way to adapt model complexity to unknown smoothness without introducing additional societal, ethical, or safety concerns beyond those standard in Bayesian optimization.

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

# A. Appendix: Theoretical Analysis and Proofs

In this appendix, we provide detailed proofs for the spectral properties of the proposed Additive Learnable $\alpha$-Stable (ALAS) Kernels. We establish the connection between the tail behavior of the spectral density and the decay rate of the Mercer eigenvalues, which ultimately governs the information gain and regret bounds in Bayesian Optimization.

## A.1. Spectral Tail and Eigenvalue Decay of the 1D ALAS Component

We use the same $2\pi$-Fourier convention as in Section 4, $k(\tau) = \int_{\mathbb{R}} e^{i2\pi\omega\tau} S(\omega)\, d\omega$. This choice only affects constant factors and does not change the asymptotic tail or eigenvalue rates.

**Proposition A.1** (Tail of symmetric $\alpha$-stable density). *Let $p(\omega; \alpha, \delta)$ be the density of a symmetric $\alpha$-stable distribution with $\alpha \in (0, 2)$ and scale $\delta > 0$, parameterized by the characteristic function $\varphi(t) = \exp(-|\delta t|^{\alpha})$. Then there exist constants $c_1, c_2 > 0$ and $\omega_0 > 0$ such that for all $|\omega| \geq \omega_0$,*

$$c_1 |\omega|^{-(1+\alpha)} \;\leq\; p(\omega; \alpha, \delta) \;\leq\; c_2 |\omega|^{-(1+\alpha)}. \tag{23}$$

*In particular, $p(\cdot; \alpha, \delta)$ is regularly varying at infinity with $-(1 + \alpha)$.*

*Proof.* This is a standard property of symmetric $\alpha$-stable laws (see (Samorodnitsky & Taqqu, 1994)). At the Gaussian boundary $\alpha = 2$, $p(\omega; 2, \delta)$ is Gaussian and decays as $\exp(-c\omega^2)$ for some $c > 0$, in contrast to the polynomial tail in (23).

**Lemma A.2** (Dominance of the heaviest tail). *Let*

$$S(\omega) = \sum_{j=1}^{J} w_j \Big[ p(\omega - \gamma_j; \alpha_j, \delta_j) + p(\omega + \gamma_j; \alpha_j, \delta_j) \Big], \qquad w_j \geq 0, \tag{24}$$

*and assume $\sum_{j=1}^{J} w_j > 0$. Define $\alpha_{\min} := \min\{\alpha_j : w_j > 0\}$. If $\alpha_{\min} < 2$, then there exist constants $C_1, C_2 > 0$ and $\omega_0 > 0$ such that for all $|\omega| \geq \omega_0$,*

$$C_1 |\omega|^{-(1+\alpha_{\min})} \leq S(\omega) \leq C_2 |\omega|^{-(1+\alpha_{\min})}. \tag{25}$$

Notably, the 1D ALAS component in (13) is a special case of the above form with $J = 1$ and $w_1 = w/2$, so the tail exponent is governed by its $\alpha$.

*Proof.* Fix any component $j$ with $w_j > 0$. By Proposition A.1, there exist constants $a_j, b_j > 0$ and $\omega_{0,j} > 0$ such that for all $|u| \geq \omega_{0,j}$,

$$a_j |u|^{-(1+\alpha_j)} \leq p(u; \alpha_j, \delta_j) \leq b_j |u|^{-(1+\alpha_j)}. \tag{26}$$

Choose

$$\omega_0 \geq \max_j \big( \omega_{0,j} + |\gamma_j| \big). \tag{27}$$

Then for all $|\omega| \geq \omega_0$ we have:

$$|\omega| - |\gamma_j| \leq |\omega \pm \gamma_j| \leq |\omega| + |\gamma_j|, \tag{28}$$

which implies $|\omega \pm \gamma_j| \asymp |\omega|$. Combining (26) and (28) yields

$$p(\omega \pm \gamma_j; \alpha_j, \delta_j) = \Theta\Big( |\omega|^{-(1+\alpha_j)} \Big). \tag{29}$$

**Upper bound.** Since $\alpha_j \geq \alpha_{\min}$ for every active component,

$$S(\omega) \leq \sum_{j:w_j>0} 2w_j\, b_j\, |\omega|^{-(1+\alpha_j)}$$

$$\leq \Big( \sum_{j:w_j>0} 2w_j b_j \Big) |\omega|^{-(1+\alpha_{\min})} =: C_2 |\omega|^{-(1+\alpha_{\min})}. \tag{30}$$

**Lower bound.** Consider the components with $\alpha_j = \alpha_{\min}$ and $w_j > 0$. Then

$$S(\omega) \geq \sum_{j:w_j > 0,\ \alpha_j = \alpha_{\min}} 2 w_j\, a_j\, |\omega|^{-(1+\alpha_{\min})}$$

$$=: C_1\, |\omega|^{-(1+\alpha_{\min})}, \tag{31}$$

where $C_1 > 0$ because at least one such component exists by definition of $\alpha_{\min}$. $\square$

**Proposition A.3** (1D eigenvalue decay). *Let $\mathcal{X} = [0, L] \subset \mathbb{R}$ and let $k(\tau)$ be a continuous stationary kernel on $\mathcal{X}$ with spectral density $S(\omega)$. We take $\nu$ to be the Lebesgue measure on $[0, L]$. Consider the Mercer operator*

$$(Tf)(x) = \int_{\mathcal{X}} k(x - x') f(x')\, dx' \quad \text{on } L_2(\mathcal{X}). \tag{32}$$

*Assume $S(\omega)$ satisfies (25) for some $\alpha_{\min} \in (0, 2)$. Then its eigenvalues $\{\lambda_h\}_{h \geq 1}$ satisfy*

$$\lambda_h = \Theta\!\left(h^{-(1+\alpha_{\min})}\right) \qquad (h \to \infty). \tag{33}$$

Here we focus on $\alpha \in (0, 2)$, where the spectrum has polynomial tails. The Gaussian boundary $\alpha = 2$ yields super-exponential decay and leads to poly-logarithmic information gain, as discussed in the main text.

*Proof.* Let $N(\varepsilon) := \#\{h : \lambda_h > \varepsilon\}$ be the eigenvalue counting function. For stationary kernels on an interval, Widom's theorem (Widom, 1963) implies that as $\varepsilon \downarrow 0$,

$$N(\varepsilon) \asymp \left|\{\omega \in \mathbb{R} : S(\omega) > \varepsilon\}\right|, \tag{34}$$

where $|\cdot|$ denotes the (1D) Lebesgue measure. The constants in $\asymp$ depend on $L$ and the Fourier convention but not on $\varepsilon$. Boundary effects are therefore absorbed into constants, while the decay exponent is governed by the spectral tail.

By (25), there exist $C_1, C_2 > 0$ and $\omega_0 > 0$ such that for all $|\omega| \geq \omega_0$,

$$C_1 |\omega|^{-(1+\alpha_{\min})} \leq S(\omega) \leq C_2 |\omega|^{-(1+\alpha_{\min})}. \tag{35}$$

Fix $\varepsilon$ small so that $R_2(\varepsilon) := (C_2/\varepsilon)^{\frac{1}{1+\alpha_{\min}}} \geq \omega_0$. Then $S(\omega) > \varepsilon$ implies $|\omega| \leq R_2(\varepsilon)$ up to the finite region $|\omega| < \omega_0$, hence

$$\left|\{\omega : S(\omega) > \varepsilon\}\right| = \mathcal{O}\!\left(\varepsilon^{-\frac{1}{1+\alpha_{\min}}}\right). \tag{36}$$

Similarly, for $R_1(\varepsilon) := (C_1/\varepsilon)^{\frac{1}{1+\alpha_{\min}}} \geq \omega_0$, the lower bound in (35) gives $S(\omega) > \varepsilon$ for $\omega_0 \leq |\omega| \leq R_1(\varepsilon)$, so

$$\left|\{\omega : S(\omega) > \varepsilon\}\right| = \Omega\!\left(\varepsilon^{-\frac{1}{1+\alpha_{\min}}}\right). \tag{37}$$

Combining (34)–(37) yields

$$N(\varepsilon) = \Theta\!\left(\varepsilon^{-\rho}\right), \qquad \rho := \frac{1}{1 + \alpha_{\min}}. \tag{38}$$

Finally, use the identity

$$\lambda_h = \sup\{\varepsilon > 0 :\ N(\varepsilon) \geq h\}. \tag{39}$$

Since $N(\varepsilon) = \Theta(\varepsilon^{-\rho})$, there exist constants $0 < c < C$ such that for all small $\varepsilon$, $c\varepsilon^{-\rho} \leq N(\varepsilon) \leq C\varepsilon^{-\rho}$. Setting $\varepsilon = (C/h)^{1/\rho}$ yields $N(\varepsilon) \leq h$, hence $\lambda_h \leq (C/h)^{1/\rho}$; setting $\varepsilon = (c/h)^{1/\rho}$ yields $N(\varepsilon) \geq h$, hence $\lambda_h \geq (c/h)^{1/\rho}$. Therefore $\lambda_h = \Theta(h^{-1/\rho}) = \Theta(h^{-(1+\alpha_{\min})})$. $\square$

### A.2. Information Gain and Regret Bounds for 1D ALAS Component

This subsection proves Theorem 4.2. The key input is Proposition A.3, which transfers the $\alpha$-stable spectral tail of the 1D ALAS component to the Mercer decay $\lambda_h = \Theta(h^{-(1+\alpha)})$. The resulting information-gain and regret bounds follow from standard GP-UCB arguments (Srinivas et al., 2009; 2012).

**Eigen-decay assumption.** For the 1D ALAS component $k_{\text{ALAS}}^{(1)}$, we have a single tail parameter, so $\alpha_{\min} = \alpha$. By Proposition A.3, the Mercer eigenvalues satisfy $\lambda_h = \Theta(h^{-\beta})$ with

$$\beta := 1 + \alpha, \qquad \alpha \in (0, 2), \tag{40}$$

so $\beta \in (1, 3)$, here $\beta$ denotes the eigenvalue-decay exponent. In particular, there exists a constant $C > 0$ such that

$$\lambda_h \le C h^{-\beta} \qquad \text{for all } h \ge 1. \tag{41}$$

**Information gain.** A standard eigenvalue bound for GP mutual information (Srinivas et al., 2009) gives

$$\gamma_T \le \frac{1}{2} \sum_{h=1}^{T} \log\left(1 + \sigma^{-2} T \lambda_h\right) \le \frac{1}{2} \sum_{h=1}^{T} \log\left(1 + cT h^{-\beta}\right), \tag{42}$$

where $c := \sigma^{-2} C$. This follows from the standard operator-to-matrix eigenvalue comparison: for any $A \subset \mathcal{X}$ with $|A| = T$,

$$\log \det(I + \sigma^{-2} K_A) \le \sum_{h=1}^{T} \log\left(1 + \sigma^{-2} T \lambda_h\right), \tag{43}$$

where $\{\lambda_h\}$ are the Mercer eigenvalues of $T_k$ on $L_2(\mathcal{X})$. Define

$$G_T := \sum_{h=1}^{T} \log\left(1 + cT h^{-\beta}\right). \tag{44}$$

Then we have $\gamma_T \le \frac{1}{2} G_T$. We bound $G_T$ by splitting the sum at the index where $cTh^{-\beta}$ is of order one. Let

$$H := \min\left\{T, \left\lceil (cT)^{1/\beta} \right\rceil\right\}. \tag{45}$$

Then for $h \le H$ we have $cTh^{-\beta} \ge 1$, and for $h > H$ we have $cTh^{-\beta} \le 1$.

**Head ($h \le H$).** For $h \le H$, $\log(1 + x) \le \log(2x)$ gives

$$\sum_{h=1}^{H} \log\left(1 + cT h^{-\beta}\right) \le \sum_{h=1}^{H} \left(\log(2cT) - \beta \log h\right)$$

$$\le H \log(2cT) = \tilde{\mathcal{O}}(H). \tag{46}$$

**Tail ($h > H$).** For $h > H$, $\log(1 + x) \le x$ gives

$$\sum_{h=H+1}^{T} \log\left(1 + cT h^{-\beta}\right) \le cT \sum_{h=H+1}^{T} h^{-\beta} \le cT \int_{H}^{\infty} x^{-\beta} \, dx$$

$$= \frac{cT}{\beta - 1} H^{1-\beta} = \mathcal{O}(T^{1/\beta}). \tag{47}$$

Combining (46) and (47) with $H = \Theta(T^{1/\beta})$ yields $G_T = \tilde{\mathcal{O}}(T^{1/\beta})$, hence

$$\gamma_T = \tilde{\mathcal{O}}(T^{1/\beta}) = \tilde{\mathcal{O}}\left(T^{\frac{1}{1+\alpha}}\right), \tag{48}$$

which proves the information-gain in Theorem 4.2.

**Regret.** Under the standard GP-UCB analysis (Srinivas et al., 2009), the cumulative regret satisfies

$$R_T = \tilde{\mathcal{O}}\left(\sqrt{T \gamma_T}\right). \tag{49}$$

Substituting (48) gives

$$R_T = \tilde{\mathcal{O}}\left(T^{\frac{1}{2} + \frac{1}{2(1+\alpha)}}\right), \tag{50}$$

which completes the proof of Theorem 4.2.

### A.3. Mercer Spectrum and Eigenvalue Decay of ALAS-Sep

We justify the spectral decomposition used for the information-gain bound of ALAS-Sep in Section 4.3. Throughout, let $\mathcal{X} = \prod_{j=1}^{d} \mathcal{X}_j$ and $\mu = \bigotimes_{j=1}^{d} \mu_j$, where each $\mu_j$ is a probability measure on $\mathcal{X}_j$.

**Centering.** For each 1D component $k^{(j)}$ on $(\mathcal{X}_j, \mu_j)$, we work with its centered version so that

$$\int_{\mathcal{X}_j} k^{(j)}(x_j, y_j)\, d\mu_j(y_j) = 0 \qquad \forall x_j \in \mathcal{X}_j. \tag{51}$$

Equivalently, the associated operator $T_j$ maps the constant function to zero. Centering here only affects finitely many eigenvalues and does not change asymptotic eigenvalue decay rates.

**Proposition A.4** (Spectrum of additive kernels). *Let the ALAS-Sep kernel be*

$$k_{ALAS}^{sep}(\mathbf{x}, \mathbf{x}') = \sum_{j=1}^{d} k^{(j)}(x_j, x_j'), \tag{52}$$

*where $k^{(j)}$ denotes the centered version of $k_{ALAS}^{(j)}$ satisfying (51). Let $T$ be the Mercer operator of $k_{ALAS}^{sep}$ on $L_2(\mathcal{X}, \mu)$, and $T_j$ be the Mercer operator of the centered $k_{ALAS}^{(j)}$ on $L_2(\mathcal{X}_j, \mu_j)$. Then every nonzero eigenvalue of $T$ is an eigenvalue of some $T_j$, and the nonzero spectrum of $T$ is the multiset union of the nonzero spectra of $\{T_j\}_{j=1}^{d}$.*

*Proof.* In this proof, we let $k^{(j)}$ denote the centered version of $k_{ALAS}^{(j)}$, so that (51) holds. Fix $j \in [d]$ and let $(\lambda_{j,m}, \phi_{j,m})$ be an eigenpair of $T_j$ with $\lambda_{j,m} \neq 0$:

$$\int_{\mathcal{X}_j} k^{(j)}(x_j, x_j')\, \phi_{j,m}(x_j')\, d\mu_j(x_j') = \lambda_{j,m} \phi_{j,m}(x_j). \tag{53}$$

Lift it to $\mathcal{X}$ by $\Psi_{j,m}(\mathbf{x}) := \phi_{j,m}(x_j)$.

**Component eigenvalues are eigenvalues of $T$** Let $k := k_{ALAS}^{sep}$. Using the additive form $k = \sum_{p=1}^{d} k^{(p)}$ and Fubini's theorem,

$$(T\Psi_{j,m})(\mathbf{x}) = \sum_{p=1}^{d} \int_{\mathcal{X}} k^{(p)}(x_p, x_p')\, \phi_{j,m}(x_j')\, d\mu(\mathbf{x}'). \tag{54}$$

The $p = j$ term reduces to (53) and equals $\lambda_{j,m} \Psi_{j,m}(\mathbf{x})$. For $p \neq j$, the integral separates into a $p$-part and a $j$-part. By centering $\int_{\mathcal{X}_p} k^{(p)}(x_p, x_p')\, d\mu_p(x_p') = 0$, so all cross terms are zero. Therefore $T\Psi_{j,m} = \lambda_{j,m}\Psi_{j,m}$.

**$T$ has no other nonzero eigenvalues.** For any $f \in L_2(\mathcal{X}, \mu)$ define the marginal

$$f_j(x_j') := \int_{\mathcal{X}_{-j}} f(\mathbf{x}')\, d\mu_{-j}(\mathbf{x}_{-j}'). \tag{55}$$

Then additivity and Fubini give the decomposition

$$(Tf)(\mathbf{x}) = \sum_{j=1}^{d} \int_{\mathcal{X}_j} k^{(j)}(x_j, x_j')\, f_j(x_j')\, d\mu_j(x_j') = \sum_{j=1}^{d} (T_j f_j)(x_j). \tag{56}$$

So $Tf$ is always a sum of $d$ one-dimensional functions. Therefore any eigenfunction $\psi$ with eigenvalue $\lambda \neq 0$ satisfies $\psi = \lambda^{-1} T\psi$ and must also be a sum of 1D functions. Therefore, every nonzero eigenvalue of $T$ must come from some $T_j$.

Combining the two steps, we show that the nonzero spectrum of $T$ is exactly the multiset union of the nonzero spectra of $\{T_j\}_{j=1}^{d}$. $\qquad \square$

**Corollary A.5** (Eigenvalue decay of ALAS-Sep). *Assume each 1D component kernel $k^{(j)}$ has Mercer eigenvalues*

$$\lambda_{j,n} = \Theta(n^{-p_j}), \qquad p_j > 1. \tag{57}$$

*Let $p_{\min} := \min_{j \in [d]} p_j$, and let $\{\lambda_h\}_{h \geq 1}$ be the sorted nonzero eigenvalues of the additive kernel (52). Then*

$$\lambda_h = \Theta\big(h^{-p_{\min}}\big). \tag{58}$$

*In ALAS-Sep, Proposition A.3 implies $p_j = 1 + \alpha_j$ with $\alpha_j \in (0, 2)$.*

*Proof.* For $\varepsilon > 0$, define the counting function

$$N_j(\varepsilon) := \#\{n : \lambda_{j,n} > \varepsilon\}. \tag{59}$$

From (57), there exist constants $0 < c_j \leq C_j$ and $\varepsilon_0 \in (0, 1)$ such that for all $\varepsilon \in (0, \varepsilon_0)$,

$$c_j\, \varepsilon^{-1/p_j} \leq N_j(\varepsilon) \leq C_j\, \varepsilon^{-1/p_j}. \tag{60}$$

By Proposition A.4, the nonzero spectrum of the sum kernel is the multiset union of the component spectra. Therefore the total counting function

$$N(\varepsilon) := \#\{h : \lambda_h > \varepsilon\} \tag{61}$$

satisfies $N(\varepsilon) = \sum_{j=1}^{d} N_j(\varepsilon)$. Let $p_{\min} = \min_j p_j$ and choose $j^\star$ with $p_{j^\star} = p_{\min}$. For $\varepsilon \in (0, 1)$ and $p_j \geq p_{\min}$, $\varepsilon^{-1/p_j} \leq \varepsilon^{-1/p_{\min}}$. Therefore, for all sufficiently small $\varepsilon$,

$$c_{j^\star}\, \varepsilon^{-1/p_{\min}} \leq N(\varepsilon) \leq \Big(\sum_{j=1}^{d} C_j\Big)\varepsilon^{-1/p_{\min}}, \tag{62}$$

so $N(\varepsilon) = \Theta(\varepsilon^{-1/p_{\min}})$. Combining $\lambda_h = \sup\{\varepsilon > 0 : N(\varepsilon) \geq h\}$ and (62), there exist constants $0 < a \leq A$ such that for all large $h$,

$$a\, h^{-p_{\min}} \leq \lambda_h \leq A\, h^{-p_{\min}}. \tag{63}$$

$\square$

### A.4. Information Gain and Regret for ALAS-Sep

**Corollary A.6** (Information gain of ALAS-Sep). *Let $k_{ALAS}^{sep} = \sum_{j=1}^{d} k^{(j)}$ be the additive kernel in (52), and assume each component has eigenvalue decay $\lambda_{j,n} = \Theta(n^{-(1+\alpha_j)})$ with $\alpha_j \in (0, 2)$. Let $\alpha_{\min} = \min_{j \in [d]} \alpha_j$. Then the maximum information gain satisfies*

$$\gamma_T(k_{ALAS}^{sep}) = \tilde{\mathcal{O}}\Big(d \cdot T^{\frac{1}{1+\alpha_{\min}}}\Big). \tag{64}$$

*Consequently, under standard GP-UCB analysis,*

$$R_T = \tilde{\mathcal{O}}\Big(\sqrt{T\, \gamma_T(k_{ALAS}^{sep})}\Big) = \tilde{\mathcal{O}}\Big(\sqrt{d}\, T^{\frac{1}{2} + \frac{1}{2(1+\alpha_{\min})}}\Big). \tag{65}$$

*Proof.* By Proposition A.4, the nonzero eigenvalues of the additive Mercer operator are the multiset union of $\{\lambda_{j,n}\}_{j,n}$. Let $\{\lambda_h\}_{h \geq 1}$ denote these eigenvalues sorted in non-increasing order.

A standard eigenvalue bound for GP information yields

$$\gamma_T(k_{ALAS}^{sep}) \leq \frac{1}{2} \sum_{h=1}^{T} \log\Big(1 + \sigma^{-2} T\, \lambda_h\Big). \tag{66}$$

**Key containment.** For each component $j$, any eigenvalue $\lambda_{j,n}$ with $n > T$ cannot be among the global top-$T$ eigenvalues, because the same component already contains $T$ eigenvalues $\{\lambda_{j,1}, \ldots, \lambda_{j,T}\}$ that are at least as large. Therefore, the global top-$T$ eigenvalues are contained in the multiset $\bigcup_{j=1}^{d}\{\lambda_{j,1}, \ldots, \lambda_{j,T}\}$.

Since $\log(1 + \sigma^{-2} T \lambda)$ is nonnegative and increasing in $\lambda$, we can upper bound the sum over the top-$T$ eigenvalues by the sum over this superset:

$$\sum_{h=1}^{T} \log\left(1 + \sigma^{-2} T \lambda_h\right) \leq \sum_{j=1}^{d} \sum_{n=1}^{T} \log\left(1 + \sigma^{-2} T \lambda_{j,n}\right). \tag{67}$$

Fix a component $j$, use $\lambda_{j,n} \leq C_j n^{-(1+\alpha_j)}$ for some $C_j > 0$ and apply the same head and tail argument as in Subsection A.2, we have

$$\sum_{n=1}^{T} \log\left(1 + \sigma^{-2} T \lambda_{j,n}\right) = \tilde{\mathcal{O}}\left(T^{\frac{1}{1+\alpha_j}}\right). \tag{68}$$

Combining (66)–(68) gives

$$\gamma_T(k_{\text{ALAS}}^{\text{sep}}) = \tilde{\mathcal{O}}\left(\sum_{j=1}^{d} T^{\frac{1}{1+\alpha_j}}\right). \tag{69}$$

Finally, since $\alpha_j \geq \alpha_{\min}$,

$$\gamma_T(k_{\text{ALAS}}^{\text{sep}}) = \tilde{\mathcal{O}}\left(d \cdot T^{\frac{1}{1+\alpha_{\min}}}\right). \tag{70}$$

The regret bound follows from the standard GP-UCB reduction $R_T = \tilde{\mathcal{O}}(\sqrt{T \gamma_T})$. $\qquad\square$

## B. Experiments

In this section, we provide a comprehensive description of our experimental setup, implementation choices, baseline methods, and additional results to ensure reproducibility.

### B.1. Implementation Details

**Code availability.** The code release is available at `https://github.com/FrankHuang24/ALAS-BO`.

**GP Modeling Framework.** All Gaussian process models are implemented in GPyTorch, with BoTorch providing the Bayesian Optimization loop. For numerical stability in exact MLL optimization and GP linear algebra (e.g., Cholesky decomposition), we use double precision for GP training and kernel computations. Observed outcomes are standardized to zero mean and unit variance before fitting. For stationary baselines (RBF, Matérn-5/2, RQ), we use the ARD variants to ensure a fair comparison with the dimension-wise parameterization used by ALAS.

**ALAS kernel construction.** Algorithm 2 summarizes how we instantiate the two kernel forms, ALAS and ALAS-Sep, from the same $\alpha$-stable spectral component. The FFT-based steps are only used to obtain a reasonable initialization from the initial design set $\mathcal{D}_{n_0}$ via an empirical spectrum heuristic. During Bayesian optimization, all kernel hyperparameters (stability, spectral, scale and noise parameters) are subsequently refined by maximizing the exact marginal log-likelihood under the same training protocol as the baselines.

**Empirical-Spectrum Initialization.** Optimizing spectral kernels is non-convex and sensitive to initialization. We use an empirical-spectrum heuristic inspired by Wilson & Adams (2013) to initialize the modulation and scale parameters. Specifically, we estimate dominant spectral frequencies from the initial observations and use them to initialize the modulation parameters ($\gamma$ for ALAS, and $\{\gamma_j\}_{j=1}^{d}$ for ALAS-Sep), together with the corresponding spectral scales ($\{\delta_j\}_{j=1}^{d}$). We initialize the stability index close to the Gaussian boundary ($\alpha = 2$ for ALAS, and $\alpha_j = 2$ for ALAS-Sep), so that marginal likelihood moves it toward rougher regimes only when supported by data.

**Hyperparameter Optimization.** All kernel hyperparameters (kernel weights, modulation, spectral scale, and stability parameter) as well as the observation noise are learned by maximizing the exact marginal log-likelihood (MLL). We enforce

---

**Algorithm 2** Construction of ALAS Kernels

---

1: **Input:** Dimension $d$, Mode $\in$ {ALAS, ALAS-Sep}
2: **Output:** Kernel function $K(\mathbf{x}, \mathbf{x}')$
3: **if** Mode is ALAS **then**
4:     **Model cross-dimensional correlations (ARD)**
5:     Initialize spectral params $\boldsymbol{\gamma}, \boldsymbol{\delta} \in \mathbb{R}^d$ via FFT on initial observations $\mathcal{D}_{n_0}$
6:     Initialize shared stability index $\alpha \in (0, 2]$
7:     **Return** $K(\mathbf{x}, \mathbf{x}') = w \exp\left(- \sum_{j=1}^d (2\pi\delta_j|\tau_j|)^\alpha\right) \cos\left(2\pi\,\boldsymbol{\gamma}^\top \tau\right)$
8: **else**
9:     **Scalable structure via decomposition**
10:     **for** dimension $j = 1$ to $d$ **do**
11:         Initialize 1D spectral params $\gamma_j, \delta_j \in \mathbb{R}$ via FFT on dim $j$
12:         Initialize dimension-specific stability index $\alpha_j \in (0, 2]$
13:         Construct component $K_j(x_j, x_j')$ using $\alpha_j, \gamma_j, \delta_j$
14:     **end for**
15:     **Return** $K(\mathbf{x}, \mathbf{x}') = \sum_{j=1}^d K_j(x_j, x_j')$
16: **end if**

---

positivity for scale and noise parameters, and keep $\alpha \in (0, 2]$ using a smooth bounded transform. We optimize the MLL with L-BFGS using the same training protocol across ALAS, ALAS-Sep, and all baselines (Zhu et al., 1997).

**Acquisition Function Optimization.** We use Expected Improvement (EI), Probability of Improvement (PI), and Upper Confidence Bound (UCB) as acquisition functions. For UCB, we fixed the exploration parameter $\beta = 0.2$ across all tasks and methods so that acquisition settings are identical. At each BO iteration, we optimize the acquisition function via multi-start gradient-based optimization: we draw $N_{\text{raw}} = 512$ raw samples within the box constraints, select $N_{\text{start}} = 10$ starting points, and run local optimization from each start. The best solution among the restarts is returned as the next candidate.

**Performance Metric.** To evaluate optimization performance consistently, we report the Log Optimality Gap at each BO iteration $t$. Let $f_t^{\text{best}}$ be the best value observed up to iteration $t$, and $f_{\text{opt}}$ be the global optimum for synthetic benchmarks. We define:

$$\text{LogGap}_t = \log_e\left(|f_t^{\text{best}} - f_{\text{opt}}|\right) \tag{71}$$

For synthetic functions, $f_{\text{opt}}$ is known analytically. For the Robot Pushing task, the objective is the distance to a target, so $f_{\text{opt}} = 0$. For Portfolio Optimization, we set $f_{\text{opt}}$ to an empirical upper bound derived from extensive offline sampling of the surrogate landscape. All experiments were repeated over 10 independent random seeds.

**Computational Resources.** All experiments were conducted on a Linux server with ARM64 CPUs (Huawei Kunpeng 920; $2\times64$ cores with SMT, 256 logical CPUs; max frequency 2.9 GHz). All methods use exact GP inference, so they share the same cubic scaling in the number of observations. The additional cost of ALAS comes from optimizing a small number of extra kernel hyperparameters.

### B.2. Baseline Kernels

We compare ALAS against commonly used stationary kernels and representative spectral baselines. All baselines are trained as exact GPs by maximizing the marginal log-likelihood, and share the same BO loop, acquisition optimization, and initialization protocol as ALAS.

**Stationary kernels.** We include RBF, Matérn-5/2, and Rational Quadratic (RQ) kernels as standard baselines. We use Matérn-5/2 as the representative Matérn baseline in the benchmark figures to keep the multi-panel plots readable, and use ARD lengthscales for all stationary baselines. For the 1D Weierstrass diagnostic in Appendix B.4, we additionally report Matérn-3/2 and Matérn-1/2, with Matérn-1/2 serving as a strong rough-kernel reference.

**Spectral kernels.** We include the Sinc kernel and the Spectral Delta Kernel (SDK) using their standard parameterizations (Tobar, 2019; Vargas-Hernández & Gardner, 2021). Their hyperparameters are learned by MLL under the same

*Table 2.* Summary of the benchmark problems. For synthetic functions, detailed definitions follow (Surjanovic & Bingham, 2013).

| TASK | GOAL | DIMENSION | ITERATIONS | INPUT DOMAIN |
|------|------|-----------|------------|--------------|
| HARTMANN | MIN | 3 | 30 | $x \in [0, 1]^3$ |
| WEIERSTRASS | MIN | 3 | 30 | $x \in [-5, 5]^3$ |
| EXPONENTIAL | MIN | 5 | 60 | $x \in [-5.12, 5.12]^5$ |
| HARTMANN | MIN | 6 | 80 | $x \in [0, 1]^6$ |
| ROSENBROCK | MIN | 10 | 150 | $x \in [-2.048, 2.048]^{10}$ |
| LEVY | MIN | 20 | 200 | $x \in [-5, 5]^{20}$ |
| RASTRIGIN | MIN | 30 | 200 | $x \in [-5.12, 5.12]^{30}$ |
| ROBOT PUSHING | MIN | 4 | 150 | $r_x, r_y \in [-5, 5]$, $t \in [1, 30], \theta \in [0, 2\pi]$ |
| PORTFOLIO | MAX | 5 | 200 | $\Omega \in [0.1, 1000], \alpha \in [5.5, 8.0]$, $C_{\text{HOLD}} \in [0.1, 100], C_{\text{BORROW}} \in [10^{-4}, 10^{-3}]$, SPREAD $\in [10^{-4}, 10^{-2}]$ |

optimization and numerical settings as ALAS.

**Remark GSM.** We do not include Gaussian Spectral Mixtures to keep structural complexity comparable: GSM is commonly instantiated with multiple Gaussian components, whereas our main ALAS models use a single $\alpha$-stable component (per dimension for the separable variant) and learn the spectral tail decay directly through $\alpha$. Extending ALAS to multi-component mixtures is a natural direction for future work.

### B.3. Test Functions

We evaluate ALAS on a set of standard BO benchmarks covering diverse input dimensions and function landscapes. The set includes smooth low-dimensional functions and more challenging multi-dimensional objectives with sharp local variation. Table 2 summarizes the test functions and their settings. For minimization benchmarks, we maximize $-f$ so that all BO loops use the same convention.

**Lower-dimensional benchmarks.** We first evaluate on lower-dimensional problems to emphasize kernel modeling behavior under limited data. This set includes smooth functions such as Hartmann ($d$=3, 6) and a low-dimensional Exponential test ($d$=5), which serve as sanity checks that ALAS can recover near-Gaussian behavior when appropriate. We also include irregular objectives such as Weierstrass ($d$=3), which is continuous but nowhere differentiable, to stress settings where learning heavier spectral tails can be beneficial.

**Larger-input benchmarks.** We further consider problems with larger input dimension to explore how the additive construction behaves. This set includes oscillatory or higher dimensional functions such as Rastrigin ($d$=30) and rugged landscapes such as Levy ($d$=20). We also include interaction-heavy objectives such as Rosenbrock ($d$=10) to examine cases where coupling across variables is important.

**Real-world problems.**

- **Robot pushing** ($d$=4). We evaluate on the Robot Pushing task, a standard real-world BO benchmark. The objective measures the final distance to a target after executing a push, given a 4-dimensional action parameterization. Since the goal is to reach a specific coordinate, the theoretical minimum distance is zero (Kaelbling & Lozano-Pérez, 2017).

- **Portfolio optimization** ($d$=5). We evaluate on a portfolio optimization task where the goal is to maximize a return-related objective by tuning a small number of decision parameters (Cakmak et al., 2020). Following prior work, we optimize a fixed surrogate objective constructed from a pre-collected set of simulator evaluations (Owen, 1998). We use the maximum value found in this offline set as the proxy for the global optimum.

### B.4. Additional Experimental Results

**Predictive-fit robustness.** Table 3 reports predictive-fit comparisons on three representative 1D tasks: Weierstrass as a rough case, Branin as a smooth case, and Rastrigin as an oscillatory case. We use the 1D versions to isolate surrogate fit

*Table 3.* Predictive fit on representative 1D tasks over $N=10$ seeds. We report test RMSE and average predictive log-likelihood (PLL).

| Kernel | Weierstrass | | Branin | | Rastrigin | |
|---|---|---|---|---|---|---|
| | RMSE $\downarrow$ | PLL $\uparrow$ | RMSE $\downarrow$ | PLL $\uparrow$ | RMSE $\downarrow$ | PLL $\uparrow$ |
| RBF | $0.31 \pm 0.11$ | $-2.86 \pm 6.07$ | $0.27 \pm 0.13$ | $0.06 \pm 0.07$ | $0.02 \pm 0.01$ | $3.18 \pm 0.11$ |
| Matérn-$1/2$ | $\mathbf{0.26 \pm 0.10}$ | $\mathbf{0.21 \pm 0.33}$ | $5.25 \pm 2.29$ | $-3.18 \pm 0.05$ | $0.79 \pm 0.17$ | $-1.35 \pm 0.02$ |
| Matérn-$3/2$ | $0.27 \pm 0.11$ | $-0.11 \pm 0.33$ | $1.25 \pm 0.64$ | $-1.11 \pm 0.08$ | $0.11 \pm 0.05$ | $0.43 \pm 0.05$ |
| Matérn-$5/2$ | $0.29 \pm 0.10$ | $-0.48 \pm 0.65$ | $0.46 \pm 0.23$ | $-0.25 \pm 0.08$ | $0.11 \pm 0.04$ | $1.60 \pm 0.07$ |
| ALAS | $\mathbf{0.26 \pm 0.09}$ | $0.01 \pm 0.63$ | $\mathbf{0.27 \pm 0.10}$ | $\mathbf{0.18 \pm 1.12}$ | $\mathbf{0.01 \pm 0.00}$ | $\mathbf{4.33 \pm 0.34}$ |

from high-dimensional search effects. These diagnostics complement the BO optimization curves by evaluating the fitted surrogate directly. Across these settings, ALAS remains competitive with standard stationary kernels while adapting its tail parameter $\alpha$ from data.

**Learned $\alpha$ and spectral decay.** We generate 1D functions with Fourier coefficients decaying as $k^{-\beta}$ and fit ALAS to each target. Figure 4 illustrates representative cases: as $\beta$ increases, the target becomes smoother and the learned $\hat{\alpha}$ moves toward the Gaussian boundary.

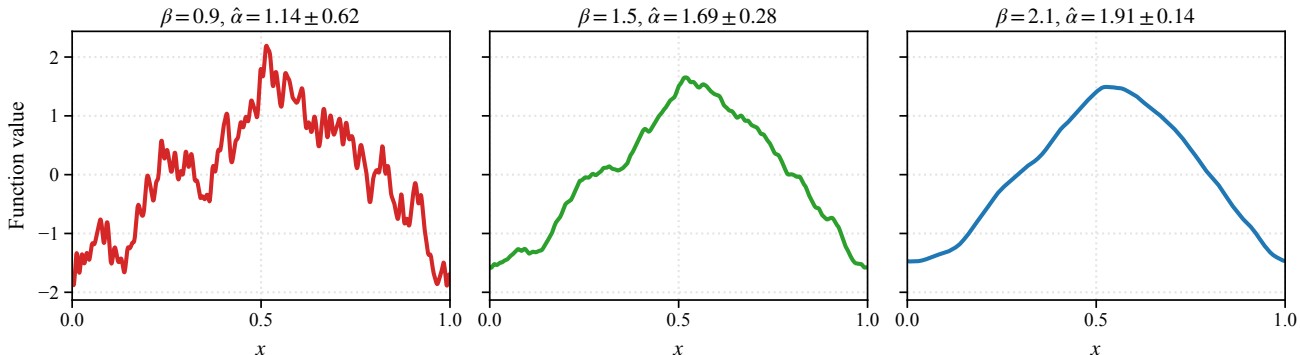

*Figure 4.* **Learned $\alpha$ tracks spectral decay.** Representative 1D functions with Fourier coefficients decaying as $k^{-\beta}$. Larger $\beta$ yields smoother targets, and the learned $\hat{\alpha}$ moves toward the Gaussian boundary $\alpha = 2$.

**Envelope-only ablation.** We compare ALAS with powered-exponential-$\alpha$ (PE-$\alpha$), which learns $\alpha$ with $\gamma = 0$. On a 1D Rastrigin, both models learn $\alpha = 2$, but ALAS learns nonzero $\gamma$ and better captures oscillation. This isolates the benefit of learning the modulation frequency beyond the powered-exponential envelope.

**Robustness across Acquisition Functions** Our main experiments use Expected Improvement (EI) for its parameter-free form and wide adoption in BO. Since different acquisition functions induce different exploration–exploitation behavior, we additionally test the robustness of ALAS by running the benchmark set with two alternative acquisitions:

- **Probability of Improvement (PI):** more exploitative than EI, favoring regions likely to improve the current best.

- **Upper Confidence Bound (UCB):** an exploration-driven alternative with an explicit parameter. Following our implementation in Appendix B.1, we fix $\beta = 0.2$ across tasks for consistency.

Overall, PI and UCB lead to trends consistent with EI across benchmarks in Fig. 3. This suggests that the gains of ALAS and ALAS-Sep mainly come from improved surrogate modeling, rather than a specific acquisition choice.

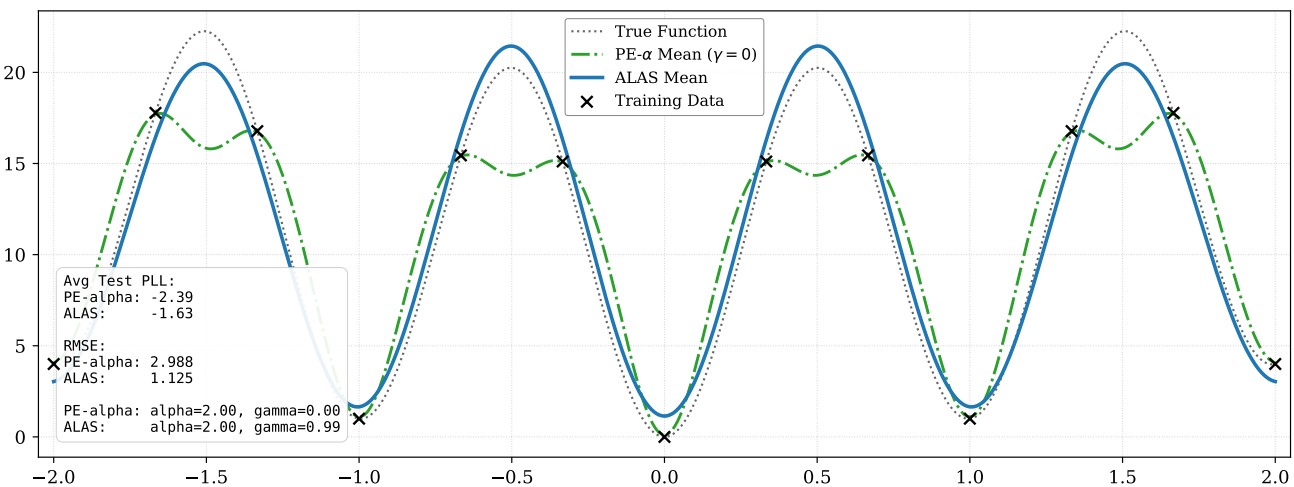

*Figure 5.* **Envelope-only ablation on 1D Rastrigin.** Powered-exponential-$\alpha$ fixes $\gamma = 0$, while ALAS also learns the modulation frequency $\gamma$. Both models learn $\alpha = 2$, but ALAS better captures the oscillatory structure.

*Figure 6.* **Optimization performance on benchmarks (PI).** Best-so-far log optimality gap over BO iterations (mean over 10 seeds) using PI.

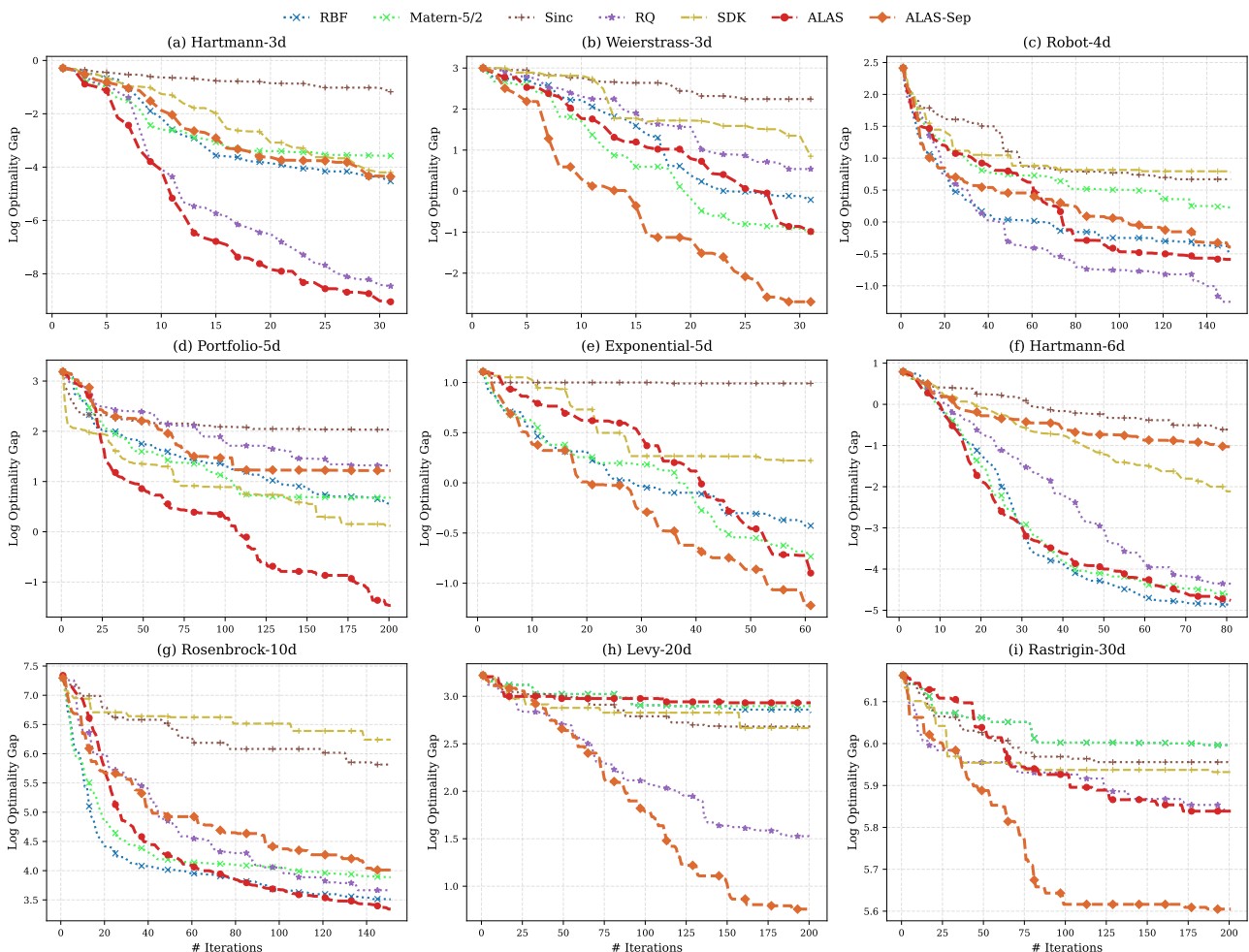

*Figure 7.* **Optimization performance on benchmarks (UCB, $\beta = 0.2$).** Best-so-far log optimality gap over BO iterations (mean over 10 seeds) using UCB.

