# OpenReview forum: "ALAS: Additive Learnable Alpha-Stable Kernels for Flexible Bayesian Optimization"
_ICML.cc/2026/Conference — ICML 2026 regular_

### Official Review · Reviewer_u1ma · 2026-02-23

**Soundness:** 4
**Presentation:** 4
**Significance:** 3
**Originality:** 3
**Overall Recommendation:** 5
**Confidence:** 4

**Summary:**

This paper proposes to use a modification of the powered-exponential kernel in GPs for Bayesian optimization, along a routine to select the specific power-exponent of the kernel. The paper includes an additive-separable version of the kernels when working with several inputs, with its corresponding routine to select the exponent. Theory-wise, the paper obtains eigenvalues through Mercer's theorem and provides bounds on the regret of the optimization scheme through said eigen values.

**Compliance With Llm Reviewing Policy:**

Affirmed.

**Final Justification:**

I recommend acceptance, the authors clarified my main concern about the Matérn baselines.

**Key Questions For Authors:**

1. If I understand correctly, the cosine component of the modulated version of the kernel (i.e., $k^{(1)}_{ALAS}$) is supposed to help when there is a periodicity component in the functional space. If so, can the authors provide any numerical experiments showing the performance of the kernel in this type of situations? Or at least indicate explicitly which of the functions contain the expected periodicity.
2. In the main text the only optimizations using Mátern kernels with roughness parameter 5/2, however in the appendix the authors include the 1/2 and 3/2 version. Notably the 1/2 performs much better than the other versions. The first thought is that the authors might be trying to hide it to highlight the improved performance of their method. However, I wonder if there is a specific reason why the authors chose to not include it in the main text.
3. Can the authors comment on the computational running time?

Smaller concerns:
- The reference in the proof of Proposition A.1 should include a page number. Probably somewhere in the first or second chapter of the book.
- The weight $w$ in Equation (12) is confusing when compared to the $\omega$ in Equation (13). I would recommend to change either of those for a smoother reading.
- Theorem 3.1, and the following examples, should include at least a reference for their respective proofs.
- The notation in Eq. (20) makes for a confusing reading with respect to Eq. (12).
- Should Line 7 of Alg. 1, start with 'obtain $y_{t+1}$' or 'collect' instead of Evaluate? If not, please provide an explanation of how it is being evaluated through the prior $f$.
- Figure 3, can the authors include some type of uncertainty quantification in the displayed graphs? This would help better ascertain that the gain is relevant.
- What domain is allowed for $\alpha$ in your implementation? All the results show $\alpha>1$, so I wonder if this a constraint or limitation of the approach, or if the current implementation could also allow $\alpha\leq 1$.
- In Eq. (51) what does the $y_j$ denote?
- Some relevant works using heavy-tailed priors, from the neural network perspective that could be relevant include: Fortuin et al (2022), Loria and Bhadra (2024, 2025), and Agapiou and Castillo.

## References:
Fortuin, V., Garriga-Alonso, A., Ober, S. W., Wenzel, F., Ratsch, G., Turner, R. E., van der Wilk, M., and Aitchison, L. (2022) Bayesian Neural Network Priors Revisited. In _International Conference on Learning Representations_

Loria, J., and Bhadra, A. (2024) Posterior Inference on Shallow Infinitely Wide Bayesian Neural Networks under Weights with Unbounded Variance. In _The 40th Conference on Uncertainty in Artificial Intelligence_

Loria, J., and Bhadra, A. (2025) Deep Kernel Posterior Learning under Infinite Variance Prior Weights. In _The Thirteenth International Conference on Learning Representations_.

Agapiou, S., and Castillo, I., (2024) Heavy-tailed Bayesian nonparametric adaptation, The Annals of Statistics, Ann. Statist. 52(4), 1433-1459

**Limitations:**

Yes

**Strengths And Weaknesses:**

## Soundness
The results appear to be technically sound. My expertise is on $\alpha$-stable random variables, and the derivations concerning these appear to be correct.
## Presentation
Presentation is good throughout. Minor concerns are mentioned below.
## Significance
Results in the main text are significant, however the Matérn(1/2) case is not included in the main paper but is present in the appendix. Leading to one of my questions below.
## Originality
The powered exponential kernel is not new and its relationship to $\alpha$-stable densities is well known, but combining it with the cosine component could be new and applying it to Bayesian optimization seems novel.

---

> ### Author Rebuttal · Authors · 2026-03-31
>
> We thank the reviewer for the careful reading and the very helpful suggestions. The main clarifications are: (i) the cosine term is intended to capture oscillatory or non-monotone structure more broadly, not only strict periodicity; (ii) Matérn-1/2 was omitted from the main text for figure clarity rather than to exclude a strong baseline, and we now make this comparison explicit; and (iii) ALAS retains exact GP inference, so its runtime has the same cubic dependence on the number of observations as other exact-GP baselines.
>
> **1. Modulation term.**
>
> We agree that this point should be made clearer. The modulation term in Eq. (12) is intended to capture oscillatory / non-monotone structure more broadly, not only strict periodicity. In the spectral domain, Eq. (13) shows that the cosine term shifts spectral mass to a symmetric pair at ±γ, while leaving the tail exponent controlled by α. To isolate this effect empirically, we added a simple ablation on a standard 1D Rastrigin task, comparing ALAS against an envelope-only variant (PE-α) that learns α but fixes γ = 0, see Fig. 1 in the [anonymous supplement](https://anonymous.4open.science/r/supp-7c65). In this experiment, both models revert to the smooth case with learned α = 2, but ALAS still learns a nonzero γ and improves clearly over PE-α. This supports the intended role of the cosine term as a mechanism for capturing oscillatory structure.
>
> **2. Matérn baseline.**
>
> We understand the concern. Matérn-1/2 was omitted from the main text for figure clarity, not to exclude a strong baseline. We originally used Matérn-5/2 as a representative because it was generally the strongest Matérn variant on many of the smoother benchmark tasks in the paper. At the same time, Matérn-1/2 is a natural rough baseline and is clearly stronger on rough objectives such as Weierstrass. We therefore make the Matérn-1/2 and Matérn-3/2 comparisons explicit in the rebuttal, both on the benchmark suite and in the added predictive-fit comparisons, see Fig. 3 in the [anonymous supplement](https://anonymous.4open.science/r/supp-7c65). For completeness, we kindly refer the reviewer to our response to Reviewer 2xK7, where we provide a more detailed predictive-fit table. More broadly, even a learned stationary Matérn ν still selects one global smoothness class. Our point is not that ALAS must dominate Matérn-1/2 on every rough task, but that it remains competitive while adapting across smooth and rough regimes without requiring the user to fix the smoothness class in advance.
>
> **3. Runtime.**
>
> ALAS keeps exact GP inference, so its runtime has the same O(n^3) dependence on the number of observations as the other exact-GP baselines. The main additional cost comes from hyperparameter optimization, and in practice fitting time can increase with input dimension as the number of kernel parameters grows. In our current experiments, this overhead remained reasonable. We will add a short runtime summary in the revision to make this point more explicit.
>
> **Minor points.**
>
> Thank you for the detailed presentation suggestions. We agree that these changes would improve readability, and we will incorporate them in the revision. Specifically, we will add a more precise reference for Proposition A.1; clean up the notation around the weights in Eqs. (12) and (13); add explicit proof or source references for Theorem 3.1 and the subsequent examples; and clarify the notation in Eqs. (20) and (51).
>
> We will also revise the wording of Algorithm 1, line 7, and clarify that this step refers to querying the true black-box objective rather than evaluating through the GP prior. We will clarify the implementation domain of α and make the reason for the constraint α ∈ (0, 2] more explicit in the revision. For a fuller explanation, we kindly refer the reviewer to our response to Reviewer Q55u. We will also add the suggested recent references on heavy-tailed Bayesian priors and adaptation. Finally, we agree that uncertainty in Fig. 3 should be presented more clearly. We omitted error bars mainly to keep the multi-panel figure readable, but we agree that showing across-seed variability would make the gains easier to assess, and we will add this in the revision.

---

> > ### Author Rebuttal · Reviewer_u1ma · 2026-04-01
> >
> > I appreciate the clarification about the periodicity. My concerns have been fully addressed, and I will update my score promptly.

---

### Official Review · Reviewer_2xK7 · 2026-02-23

**Soundness:** 3
**Presentation:** 3
**Significance:** 2
**Originality:** 3
**Overall Recommendation:** 4
**Confidence:** 4

**Summary:**

The manuscript proposes a class of new stationary kernels that promise better identification and learning of smoothness in a BO context.

**Compliance With Llm Reviewing Policy:**

Affirmed.

**Final Justification:**

After careful consideration of my initial review and the authors' response, I recommend a "weak accept". I have increased my score in response to the authors' rebuttal.

**Key Questions For Authors:**

- How does the algorithm perform in function approximation tasks compared to other stationary and non-stationary kernels, measured by RMSE, CRPS, or similar scores?
- How does the new kernel compare to learning a smoothness for Matern kernels in the stationary and non-stationary case?

**Limitations:**

No limitations should be made clearer. Especially the mentioned issue of comparing to non-stationary kernels, where the differentiability can be a function over the input space.

**Strengths And Weaknesses:**

Soundness:
Is the submission technically sound?
I am not able to identify mistakes.


Are claims well supported?
Not all, no. The biggest concern I have is the complete omission of a non-stationary kernel. Many claims that motivate this work focus only on stationary kernels. But of course, a GP that “adapts its effective smoothness from data, capturing both smooth trends and sharp irregularities” is only novel in the stationary case. Simply saying that the authors deal with the part of the GP universe that only considers stationary kernels is insufficient. Deep kernels are mentioned, but otherwise non-stationarity is ignored. In fact, there are non-stationary kernels that not only learn smoothness but also learn smoothness as a function on the input space.
See, for example, https://arxiv.org/pdf/1610.02447


Are the methods used appropriately?
Beyond the point made above, the methods are used appropriately.

If the paper includes theoretical results, are the proofs correct and based on reasonable assumptions?
Yes

Are the experiments well-designed?
No, a BO optimization success depends on two things: a good approximation of the function and an acquisition function that optimally utilizes the approximation. Therefore, to show that BO improves the algorithm's function approximation ability, this has to be carefully studied. I recommend studying and presenting RMSE, CRPS, and Probability Coverage scores for the new kernels on a variety of test functions. This will help identify the true power of the proposed kernels.

Are the authors careful and honest about evaluating both the strengths and weaknesses of their work?
No, a clear comparison has to be drawn to a non-stationary kernel.

Presentation:
Is the submission clearly written and well-structured?
Yes, the manuscript is very easy to follow.

Is the overall narrative easy to follow?
Yes.

Does the work properly position itself in the context of prior/concurrent literature and clearly discuss how it differs?
No, with the main critique that non-stationary kernels are largely omitted.

Significance:
Does the paper address an important or relevant problem?
Yes, kernel choice for GPs is important, even in the stationary case.

Does it advance understanding, capabilities, or practice in machine learning?
Yes. Although the advancement is small, given the large corpus of work on non-stationary kernels.

Could it influence future research or applications (e.g., other researchers or practitioners are likely to use the ideas or build on them)?
Yes

Is the scope of impact broad or specialized, and is that appropriate for the contribution?
The impact would be quite specialized, in particular, for users who are willing to implement a more advanced stationary kernel, but unwilling to learn differentiability in a possibly non-stationary Matern kernel.

Even if the improvements are modest or domain-specific, could they unlock new directions or provide practical utility?
Yes.


Originality:
Does the work provide new insights, deepen understanding, or highlight important properties of existing methods?
Spectral kernels are well understood, but the manuscript makes some advancements in that general field.

Does the work introduce new tasks, methods, theory, data, or perspectives that advance the field in some dimensions?
No, not currently, but this might have to do with insufficient experimental evaluation.

Does this work offer a novel combination of existing techniques, and is the reasoning behind this combination well-articulated?
Once a clear and correct explanation is provided of why a user should not just learn the differentiability hyperparameter of stationary and non-stationary  Matern kernels, yes. Currently no.

Are the contributions clearly distinguished from closely related literature, and is the novelty well justified?
No, for the reason of learned differentiability as a function of the input space in non-stationary kernels.

---

> ### Author Rebuttal · Authors · 2026-03-30
>
> We thank the reviewer for the detailed and thoughtful comments. To address the reviewer’s concerns, we added direct predictive-fit comparisons (RMSE and predictive log-likelihood) and comparisons with two representative non-stationary baselines.
>
> **1. Relation to non-stationary kernels.**
>
> We agree that non-stationary GP models can capture input-dependent smoothness, and we will revise the introduction and related work to better reflect this literature. At the same time, ALAS differs in scope: it is designed as a stationary kernel family for BO, where exact GP structure and analytical tractability remain central. To further address the reviewer’s concern, we added two representative non-stationary baselines: a non-stationary Matérn-5/2 kernel and an adaptive Gibbs kernel. The former is closely related to the Matérn-based suggestion, while the latter provides a second representative non-stationary baseline.
>
> **2. Surrogate-fit evidence.**
>
> Some surrogate-level evidence was already present in the original submission: Appendix B.4 reports predictive-fit results on Weierstrass and acquisition ablations across EI / PI / UCB. We further added predictive-fit comparisons using RMSE and predictive log-likelihood (PLL) to broaden this evidence. These new experiments extend the original evidence rather than introduce it for the first time.
>
> We now report predictive-fit comparisons on three representative tasks: Weierstrass as a rough case, Branin as a smooth case, and Rastrigin as an oscillatory case. We use the 1D versions to isolate surrogate fit more cleanly. The combined results are shown in the table below, and the full benchmark plots are provided in Fig. 3 of the [anonymous supplement](https://anonymous.4open.science/r/supp-7c65). Overall, the new results are consistent with the main paper: ALAS and ALAS-Sep remain competitive across rough, smooth, and oscillatory settings, including against the added non-stationary baselines.
>
> | Kernel        | Wei. RMSE   | Wei. PLL    | Bra. RMSE   | Bra. PLL    | Ras. RMSE   | Ras. PLL    |
> |---------------|-------------|-------------|-------------|-------------|------------|------------|
> | RBF           | 0.31±0.11   | -2.86±6.07  | **0.27±0.13** | 0.06±0.07   | 0.02±0.01  | 3.18±0.11  |
> | Matérn-1/2    | **0.26±0.10** | **0.21±0.33** | 5.25±2.29   | -3.18±0.05  | 0.79±0.17  | -1.35±0.02 |
> | Matérn-3/2    | 0.27±0.11   | -0.11±0.33  | 1.25±0.64   | -1.11±0.08  | 0.11±0.05  | 0.43±0.05  |
> | Matérn-5/2    | 0.29±0.10   | -0.48±0.65  | 0.46±0.23   | -0.25±0.08  | 0.11±0.04  | 1.60±0.07  |
> | Gibbs         | 0.29±0.09   | -1.22±1.15  | 0.37±0.16   | -0.05±0.17  | 0.04±0.02  | 2.37±0.10  |
> | NS-Matérn-5/2 | 0.30±0.13   | -0.37±0.41  | 0.46±0.23   | -0.25±0.08  | 0.11±0.04  | 1.61±0.05  |
> | ALAS          | **0.26±0.09** | 0.01±0.63   | **0.27±0.10** | **0.18±1.12** | **0.01±0.00** | **4.33±0.34** |
>
> **3. Learning Matérn smoothness.**
> We agree that learning Mat\'ern smoothness, in both stationary and non-stationary forms, is an important related direction. These correspond to different model structures. A learned stationary Mat\'ern $\nu$ still selects one global regularity class for the whole kernel (see Rasmussen and Williams, 2006, Ch. 4), whereas non-stationary Mat\'ern models are designed for input-dependent local smoothness (Paciorek and Schervish, 2003).
>
> Within the stationary setting we study, ALAS is different from simply learning a Mat\'ern $\nu$. It learns the spectral-tail parameter $\alpha$ directly and can also capture oscillatory or non-monotone structure through $\gamma$. In addition, ALAS-Sep allows different effective regularity across dimensions through per-dimension $\alpha_j$, which is a different kind of flexibility from a single global Mat\'ern smoothness parameter.
>
> For non-stationary Mat\'ern models, we agree that input-dependent smoothness is possible and important. Our point is not to claim the same advantage over that model class. We only note that learning general Mat\'ern smoothness is not always straightforward in practice, since derivatives with respect to $\nu$ can be computationally delicate (Geoga et al., 2023). Our focus here is different: ALAS stays within an exact-GP framework, while ALAS-Sep is aimed at the low-to-moderate-dimensional setting studied in the paper.
>
> **References**
>
> - Rasmussen, C. E., and Williams, C. K. (2006). Gaussian Processes for Machine Learning. *MIT Press*.
> - Paciorek, C. J., and Schervish, M. J. (2003). Nonstationary covariance functions for Gaussian process regression. *In Advances in Neural Information Processing Systems 16*.
> - Geoga, C. J., Wenger, J., and Stein, M. L. (2023). Computing Gaussian process likelihoods and their derivatives for general Matérn covariances. *Journal of Machine Learning Research, 24(141), 1–26*.

---

> > ### Author Rebuttal · Reviewer_2xK7 · 2026-03-31
> >
> > I like the changes the authors made, and I agree with the scope more so now. I will adjust my score accordingly.

---

### Official Review · Reviewer_m6fP · 2026-03-04

**Soundness:** 3
**Presentation:** 3
**Significance:** 2
**Originality:** 3
**Overall Recommendation:** 5
**Confidence:** 4

**Summary:**

This paper introduces ALAS, a novel GP kernel family for BO designed to automatically adapt to the unknown smoothness of an objective function. Instead of forcing practitioners to choose between smooth or rough kernels beforehand, the method models the kernel's spectral density using symmetric alpha-stable components. By treating the stability parameter (alpha) as a learnable hyperparameter, the kernel can smoothly interpolate between traditional, highly regular Gaussian behavior and rougher, heavy-tailed behavior. The authors propose two versions of this approach: a standard multi-dimensional version with a shared stability parameter, and a separable version (ALAS-Sep) that learns a distinct roughness parameter for each individual dimension to better handle additive structures. On the theoretical side, the paper formally connects the heaviness of the spectral tails to Mercer eigenvalue decay, establishing rigorous information gain and regret bounds that scale directly with the learned alpha parameter. The framework is evaluated on a range of synthetic and real-world optimization tasks (ranging 3-30 dims) including robot pushing and portfolio optimization, demonstrating its ability to adapt to the underlying landscape.

**Compliance With Llm Reviewing Policy:**

Affirmed.

**Final Justification:**

I think this paper should be accepted for all of the reasons indicated in my initial review. I initially gave a score of 4, and have now updated my score to 5 to reflect that the author's rebuttal addressed all of my questions and concerns.

**Key Questions For Authors:**

**Q1:** Table 1 shows alpha near 2 for smooth functions and near 1.2-1.6 for rough ones. Can you characterize the relationship between learned alpha and function regularity more precisely? For instance, how does alpha relate to Holder continuity or Fourier coefficient decay? A mapping from alpha to interpretable smoothness measures would help practitioners assess whether the fitted kernel is appropriate for their particular application.

**Q2:** How often does L-BFGS optimization fail, triggering the Adam fallback? Are there identifiable conditions (e.g., small n, extreme alpha, particular function landscapes) where alpha estimation becomes unreliable?

**Q3:** For ALAS-Sep in truly high dimensions (d=100+), does learning per-dimension alpha_j remain tractable? Have you considered grouped or hierarchical structures where subsets of dimensions share alpha?

**Limitations:**

yes

**Strengths And Weaknesses:**

### Soundness
Strengths:
Technical foundations are solid. Bochner's theorem is correctly applied with alpha-stable spectral densities, and the eigenvalue decay analysis (Proposition 4.1, Theorem 4.2) rigorously connects spectral tails to information gain. Regret bounds follow cleanly from GP-UCB analysis (Srinivas et al., 2009) combined with the new eigenvalue characterization. ALAS-Sep properly exploits additive structure to maintain exact GP inference. The experimental design is thorough, with appropriate baselines (RBF, Matern, etc.).

Weaknesses: 1) The compact domain assumption is stated, but boundary effects are not discussed. 2) The fallback optimization strategy (L-BFGS then Adam, described in Appendix B.1) points to occasional MLL instability that deserves more analysis: when does this happen and how often? 3) While learned alpha values track function regularity empirically (Table 1), there is no theoretical characterization of when alpha estimation is reliable or how many samples are needed for it to stabilize?

### Presentation
Strengths:
The paper is generally well-written. Figures 1 and 2 effectively illustrate ALAS adapting to rough versus smooth functions. The spectral density formulation and its connection to Mercer eigenvalues are clearly explained, and Algorithm 1 is easy to follow.

Weaknesses:
I found the learned alpha values in Table 1 interesting but under-explored. A more thorough discussion/analysis of what drives particular alpha values would strengthen the paper.

### Significance
Strengths:
Automatic kernel adaptation addresses a genuine practical need. Practitioners often guess between RBF, Matern, and other fixed-smoothness options, and ALAS provides a principled alternative. The theoretical contribution linking spectral tails to information gain and regret is also valuable.

Weaknesses:
The empirical gains are often modest. On smooth benchmarks (Hartmann, Exponential), ALAS essentially recovers RBF/Matern performance with no clear advantage. The wins concentrate on rough functions (e.g., Weierstrass, Rastrigin) where alpha drops below ~1.5. Furthermore, cubic inference cost limits scalability (experiments cap at $n \le 200$). The largest benchmark is $d=30$, reasonable but not highly dimensional by modern standards. Also, the paper does not provide practical guidance on when ALAS is preferable to simply trying a few standard kernels.

### Originality
Strengths:
The core idea of making alpha-stable tail exponents learnable for GP kernels is novel and well-motivated. This clearly differs from spectral mixture kernels (Wilson & Adams, 2013), which use Gaussian mixture spectral densities and cannot learn tail behavior, and from deep kernel learning (Wilson et al., 2016), which learns feature maps but keeps fixed base kernels. The eigenvalue decay and information gain analysis for alpha-stable components is novel, and ALAS-Sep's per-dimension tail learning is an original design choice.

Weaknesses:
The individual building blocks are standard: alpha-stable distributions are well-established (Samorodnitsky & Taqqu, 1994; Nolan, 2020), MLL-based hyper-parameter learning is routine, and the additive decomposition follows Kandasamy et al. (2015) and Gardner et al. (2017). The modulation mechanism in Eq. 12 also resembles existing spectral modulation approaches. However, this is a minor weakness, the ideas presented are still novel in my opinion as the novelty lies in the specific combination and the theoretical analysis, rather than in any single component.

---

> ### Author Rebuttal · Authors · 2026-03-31
>
> We thank the reviewer for the careful reading and constructive questions. In the rebuttal, we add a simple 1D calibration experiment for learned $\alpha$, clarify the L-BFGS to Adam fallback as a numerical safeguard, and discuss the practical scope of ALAS-Sep in higher dimensions.
>
> **1. Interpreting learned $\alpha$.**
>
> We agree that Table 1 is mainly qualitative, and that a more direct calibration of learned $\alpha$ is useful. We do not view $\alpha$ as a literal estimator of a classical smoothness order such as the Hölder exponent. Instead, $\alpha$ is better interpreted as an effective spectral-tail indicator: smaller $\alpha$ corresponds to heavier high-frequency content and slower eigenvalue decay, while larger $\alpha$ corresponds to smoother behavior and faster spectral decay. In the periodic setting, classical analysis shows that Hölder regularity implies polynomial decay of Fourier coefficients (see Zygmund, 2002, Sec. II.4), but this implication is one-sided rather than invertible.
>
> For this reason, rather than claiming a universal closed-form relation between $\alpha$ and Hölder continuity, we calibrate $\alpha$ directly against Fourier coefficient decay. Specifically, we added a 1D synthetic study in which the target has Fourier coefficients decaying as $k^{-\beta}$. This calibration is summarized below and shown as Fig. 2 in the [anonymous supplement](https://anonymous.4open.science/r/supp-7c65). Larger $\beta$ corresponds to faster spectral decay and a smoother target by construction. We observe a clear monotone trend: the learned $\hat{\alpha}$ increases with $\beta$ and approaches the Gaussian boundary $\alpha=2$ for the smoothest cases. We therefore view this as a practical calibration: smoother spectral decay leads to larger learned $\alpha$, consistent with our spectral-tail interpretation.
>
> | Fourier decay $\beta$ | Learned $\hat{\alpha}$ (mean $\pm$ std) |
> |---|---:|
> | 0.7 | $0.46 \pm 0.57$ |
> | 0.9 | $1.14 \pm 0.62$ |
> | 1.1 | $1.34 \pm 0.47$ |
> | 1.3 | $1.56 \pm 0.35$ |
> | 1.5 | $1.69 \pm 0.29$ |
> | 1.7 | $1.79 \pm 0.23$ |
> | 1.9 | $1.86 \pm 0.18$ |
> | 2.1 | $1.91 \pm 0.15$ |
>
> As a practical takeaway, we view ALAS as most useful when the objective smoothness is uncertain or mixed, whereas standard kernels remain strong simple choices when the objective is known to be smooth in advance.
>
> **2. Optimization stability.**
>
> The L-BFGS to Adam fallback was introduced as a robustness safeguard because we did occasionally observe optimization instability. In practice, we found this behavior to be concentrated in the earliest BO iterations and to become much less frequent as more data accumulate. Our current understanding is that this is mainly a small-$n$ numerical issue, and it can be more likely in models with more kernel parameters such as ALAS-Sep. With limited pairwise-distance information, different hyperparameter settings can induce very similar covariance matrices. In ALAS, this is amplified by the tradeoff between $\alpha$ and $\delta$, which makes the marginal log-likelihood less identifiable in the early BO stages.
>
> Adam is used only as a conservative fallback in these cases, since it tends to be more stable when the marginal-likelihood optimization is poorly conditioned. In our more recent reruns, this fallback became much less frequent after minor robustness refinements, which further suggests that the issue is mainly numerical rather than conceptual.
>
> **3. Practical dimensional structures.**
>
> For truly high-dimensional settings ($d \ge 100$), we do not view fully free per-dimension $\alpha_j$ learning in ALAS-Sep as the final answer under tight BO budgets. ALAS-Sep is intended as an exact-GP construction for settings where the objective is reasonably low or moderate-dimensional and close to additive. We agree that grouped or hierarchical sharing of $\alpha$ is a promising route in higher dimensions, since it provides a natural bridge between shared-$\alpha$ ALAS and fully per-dimension ALAS-Sep while keeping the model identifiable under limited data. We will revise the discussion to make this practical scope clearer.
>
> **References**
>
> * Zygmund, A. (2002). *Trigonometric Series*. Cambridge University Press.

---

> > ### Author Rebuttal · Reviewer_m6fP · 2026-04-03
> >
> > The authors addressed each of my concerns and answered each of my questions from my review. I will update my score accordingly.

---

### Official Review · Reviewer_Q55u · 2026-03-11

**Soundness:** 3
**Presentation:** 3
**Significance:** 2
**Originality:** 1
**Overall Recommendation:** 4
**Confidence:** 4

**Summary:**

This paper focuses on the problem of mismatch between the smoothness of the kernel function and the true objective function in Bayesian optimization. Existing commonly used stationary kernels often implicitly assume a fixed smoothness, but the regularity of the objective function in Bayesian optimization is usually unknown beforehand. Therefore, pre-selecting the smoothness of the kernel function may lead to modeling bias and decreased sample efficiency. To address this issue, this paper proposes a stationary kernel family (ALAS) based on the symmetric α-stable spectral component, where the parameter α controls the heaviness of the spectral tail, thereby adjusting the effective regularity of the kernel. The authors learn the parameters in this spectral representation from the data through GP marginal likelihood, thus obtaining a kernel that better matches the data. Furthermore, the paper proposes a separable multidimensional extension ALAS-Sep, allowing different input dimensions to learn their respective roughness parameters.

**Compliance With Llm Reviewing Policy:**

Affirmed.

**Final Justification:**

The authors addressed my concerns. I increase my score to 4.

**Key Questions For Authors:**

1. The paper sets the range of $\alpha$ to (0, 2], which is natural under the definition of the α-stable family. However, the paper does not fully explain why this range is the most meaningful and necessary choice for the kernel parameterization in this paper.
2. The paper defines ALAS starting from the spectral density and obtains the corresponding covariance through Bochner's theorem; however, the paper also clearly points out that (11) corresponds to the classical powered-exponential covariance family. Based on this, I hope the authors will further clarify why this paper is more suitable to describe as learning the $\alpha$-stable spectral density, rather than more directly as learning the powered-exponential covariance family.
3. The theoretical section presents a complete analytical chain from spectral tail behavior to eigenvalue decay, information gain, and regret, but overall it is highly consistent with the classic GP bandit/spectral-decay analysis framework. Therefore, I would like the authors to more clearly distinguish which parts are truly new theoretical contributions and which parts are mainly instantiations and adaptations of this kernel family under existing theoretical templates.

**Limitations:**

yes

**Strengths And Weaknesses:**

Strengths:
I believe the problem addressed in this paper is reasonable and practically significant: in Bayesian optimization, commonly used Gaussian process kernels often implicitly assume fixed smoothness, but the roughness of the true objective function is often unknown beforehand, thus mismatches can indeed occur between the surrogate kernel and the objective function. The proposed ALAS and its separable extension ALAS-Sep attempt to regulate spectral tail behavior and effective regularity through learnable α-stable spectral components. This modeling approach is natural within the GP framework; the theoretical and experimental sections are generally consistent with the methodological settings, and I have not seen any obvious technical errors. The paper also presents an analytical chain from spectral tail decay to Mercer eigenvalue decay, and then to the maximum information gain and regret upper bound, and covers composition functions, high-dimensional tasks, and several application scenarios in the experiments, demonstrating a satisfactory overall level of completion.

Weaknesses:
While the problem setting is reasonable and the method is largely self-consistent within the GP framework, I have concerns about the overall contribution of the paper.
1. From a significance perspective, I agree that the paper addresses a real issue in Bayesian optimization, namely the potential mismatch between a fixed kernel smoothness assumption and an unknown objective function. I also acknowledge that the proposed method may provide additional modeling flexibility in practice. However, based on the current results, I view the work primarily as an incremental extension rather than a contribution that substantially advances this research direction.
2. From an originality perspective, the core approach appears closely connected to existing powered-exponential and spectral-kernel lines of work. The spectral representation provides a natural modeling narrative and a convenient entry point for the subsequent analysis, but its methodological necessity beyond offering a spectral interpretation and an analysis route is not fully convincing. In this sense, the method comes across more as a spectral reinterpretation and reparameterization of an existing kernel family than as a clearly new methodological object.
3. The theoretical analysis is complete, but it mainly follows existing spectral analysis and GP bandit theory templates: it derives eigenvalue decay from spectral tail behavior, and then uses this to control information gain and regret. As a result, the theory is better viewed as an adaptation of an existing analytical framework to this kernel family, rather than as providing new theoretical tools or materially extending prior conclusions.
4. The experimental results suggest that the method is workable and reasonably robust across several tasks. However, they are not yet strong enough to offset the limited novelty of the method and theory, nor do they convincingly demonstrate a level of impact commensurate with acceptance at a selective venue.
Overall, I find the paper reasonable and technically sound, but its main contribution is closer to a systematic and incremental extension of existing kernel design and GP-based Bayesian optimization analysis frameworks. For this reason, I remain unconvinced about its acceptance priority, mainly due to limited originality and significance.

---

> ### Author Rebuttal · Authors · 2026-03-30
>
> We thank the reviewer for the careful reading and thoughtful comments. We focus on three points related to the reviewer’s originality concern:
> (i) why the domain $\alpha \in (0,2]$ is natural for our parameterization;
> (ii) why the paper is better understood through the spectral view than through the powered-exponential envelope alone; and
> (iii) what is new in the theory, and what follows standard GP-bandit templates.
>
> **1. The range of $\alpha$.**
>
> Thank you for raising this point. We agree that the current draft did not explain this choice clearly enough. We do not impose $(0,2]$ arbitrarily; rather, this is the natural parameter domain of the family we build on. On the spectral side, $\alpha \in (0,2]$ is exactly the classical stability-index range for symmetric $\alpha$-stable laws, with $\alpha=2$ corresponding to the Gaussian case (see Nolan, 2020, Ch. 1, p. 7). On the covariance side, the base envelope in Eq. (11) is the standard powered-exponential covariance family, which is positive definite only for $0<\alpha\le 2$ (see Rasmussen and Williams, 2006, Ch. 4). Thus, $(0,2]$ is the largest range that preserves both the $\alpha$-stable spectral interpretation and the positive-definiteness of the covariance family. We will revise the paper to make this point explicit.
>
> **2. The spectral view.**
>
> We agree that the base envelope in Eq. (11) comes from the classical powered-exponential covariance family. However, Eq. (11) is only the envelope. The ALAS component used in the paper is the modulated kernel in Eq. (12), with $(\alpha,\delta,\gamma)$ learned jointly.
>
> This is why the spectral view is more informative. In covariance form, $\alpha$ appears only as an envelope exponent. In spectral form, $\alpha$ and $\gamma$ play different roles: $\alpha$ controls the tail exponent and hence the effective regularity, while $\gamma$ shifts spectral mass away from zero and induces oscillatory or non-monotone correlations. Eq. (13) makes this explicit by showing that modulation shifts the base spectrum to a symmetric pair at $\pm \gamma$ without changing the tail exponent.
>
> We also added a simple 1D Rastrigin ablation in the rebuttal, comparing ALAS with an envelope-only variant (PE-$\alpha$) that learns $\alpha$ but fixes $\gamma=0$, see Fig. 1 in the [anonymous supplement](https://anonymous.4open.science/r/supp-7c65). In this experiment, both models learn $\alpha=2$, but ALAS still learns a nonzero modulation parameter and improves clearly over PE-$\alpha$. This shows that the benefit of ALAS is not only the powered-exponential envelope in Eq. (11), but also the shifted-spectrum flexibility introduced by $\gamma$. We will revise the paper to make this distinction more explicit.
>
> **3. The theoretical contribution.**
>
> We agree that the proof route follows the classical GP-bandit / spectral-decay framework, rather than introducing a new general proof technique. The novelty lies in the kernel family being analyzed and in the kernel-specific conclusions that follow.
>
> At the modeling level, ALAS turns fixed $\alpha$-stable forms into a learnable kernel family, so that Gaussian-like ($\alpha=2$), Cauchy-like ($\alpha=1$), Lévy-like ($\alpha=1/2$), and intermediate regimes are handled within one unified parameterization. At the theoretical level, for the one-dimensional ALAS component we derive the eigenvalue decay
>
> $$
> \lambda_h = \Theta\bigl(h^{-(1+\alpha)}\bigr)
> $$
>
> and then obtain the corresponding $\alpha$-dependent information-gain and regret bounds. For ALAS-Sep, we further extend this analysis to the additive setting, where the rate is governed by $\alpha_{\min}$.
>
> We agree that the proof template itself is standard. Our contribution lies in the learnable $\alpha$-stable kernel family and the explicit $\alpha$-dependent eigenvalue, information-gain, and regret guarantees that follow from it. We will revise the paper to make this distinction clearer.
>
> **References**
>
> * Nolan, J. P. (2020). Univariate Stable Distributions. *Springer*.
> * Rasmussen, C. E., and Williams, C. K. I. (2006). Gaussian Processes for Machine Learning. *MIT Press*.

---

> > ### Author Rebuttal · Reviewer_Q55u · 2026-04-03
> >
> > I appreciate the detailed response. I still want to better position the theoretical contribution to judge the score. Is there any technical challenges on: one-dimensional ALAS component you derive the eigenvalue decay, and the further step of obtaining the regret bounds? If so, can the authors provide with more details? The authors highlight this as part of theoretical contribution but it seems straightforward. On the other hand, as the authors mentioned in the response that they further extend to the additive setting, it is not quite clear about this ALAS-Sep part.

---

> > > ### Author Response · Authors · 2026-04-03
> > >
> > > We thank the reviewer for the helpful follow-up. We agree that once Mercer eigenvalue decay is established, the final step to information gain and regret follows the standard GP-bandit analysis. Our main theoretical contribution is to establish explicit Mercer eigenvalue decay laws for the proposed ALAS family, both for the one-dimensional ALAS component and for the additive construction ALAS-Sep.
> > >
> > > For the one-dimensional ALAS component, our theory establishes an explicit Mercer eigenvalue decay law for the modulated $\alpha$-stable family. This contrasts with many existing BO kernel analyses, where information gain is typically derived from known smoothness classes or previously established eigenvalue decay results (see Vakili et al., 2021). The key step here is to transfer the Fourier-domain $\alpha$-stable tail law to the Mercer spectrum. Using Widom's eigenvalue-counting argument, we show
> > >
> > > $$
> > > N(\varepsilon)=\Theta\left(\varepsilon^{-1/(1+\alpha)}\right), \qquad
> > > \lambda_h=\Theta\left(h^{-(1+\alpha)}\right).
> > > $$
> > >
> > > Thus, the learnable tail parameter $\alpha$ directly controls the Mercer eigenvalue decay, which yields explicit information-gain and regret bounds via standard GP-bandit analysis.
> > >
> > > For ALAS-Sep, the main contribution in the additive setting is to characterize the Mercer spectrum of the additive construction with coordinate-wise tail parameters $\alpha_j$. After centering, we show that the nonzero spectrum of the additive Mercer operator is exactly the multiset union of the one-dimensional component spectra. This yields
> > >
> > > $$
> > > \lambda_h=\Theta\left(h^{-(1+\alpha_{\min})}\right), \qquad
> > > \alpha_{\min}:=\min_{j\in[d]} \alpha_j.
> > > $$
> > >
> > > So the overall decay is governed by the least smooth coordinate, while the dimension dependence enters only through a linear factor $d$ in the information-gain bound.
> > >
> > > **References**
> > > - Vakili, S., Khezeli, K., and Picheny, V. (2021). *On Information Gain and Regret Bounds in Gaussian Process Bandits*. In AISTATS 2021.

---

### Decision · Program_Chairs · 2026-04-30

**Decision:**

Accept (regular)

**Comment:**

All reviewers found the paper technically sound. The paper makes both a theoretical contribution (a new kernel family and its analysis) and provides a practical new tool for Bayesian optimization. During the discussion, the authors clarified the novelty and scope of the method, explained the theoretical contribution more clearly, and added extra comparisons, addressing reviewers’ concerns. Some reviewers thought that the work may feel somewhat incremental and that the empirical gains are not equally strong on every task. Still, in my opinion the practical value is clear: the method matches standard kernels on smooth target functions and improves substantially on rough ones, without requiring prior knowledge that the function is rough. This is a very convenient and useful property in practice. The theory mostly relies on standard techniques, but it still provides meaningful support for this new method. Overall, I recommend acceptance.